# Cerebral malaria is associated with differential cytoadherence to brain endothelial cells

Janet Storm[1,2,3,*] (iD), Jakob S Jespersen[4,5] (iD), Karl B Seydel[3,6,7], Tadge Szestak[1], Maurice Mbewe[2], Ngawina V Chisala[2], Patricia Phula[2], Christian W Wang[4,5], Terrie E Taylor[6,7], Christopher A Moxon[8,9], Thomas Lavstsen[4,5] (iD) & Alister G Craig[1,**] (iD)

## Abstract

Sequestration of *Plasmodium falciparum*-infected erythrocytes (IE) within the brain microvasculature is a hallmark of cerebral malaria (CM). Using a microchannel flow adhesion assay with TNF-activated primary human microvascular endothelial cells, we demonstrate that IE isolated from Malawian paediatric CM cases showed increased binding to brain microvascular endothelial cells compared to IE from uncomplicated malaria (UM) cases. Further, UM isolates showed significantly greater adhesion to dermal than to brain microvascular endothelial cells. The major mediator of parasite adhesion is *P. falciparum* erythrocyte membrane protein 1, encoded by *var* genes. Higher levels of *var* gene transcripts predicted to bind host endothelial protein C receptor (EPCR) and ICAM-1 were detected in CM isolates. These data provide further evidence for differential tissue binding in severe and uncomplicated malaria syndromes, and give additional support to the hypothesis that CM pathology is based on increased cytoadherence of IE in the brain microvasculature.

**Keywords** cerebral malaria; cytoadherence; paediatric patient isolates; PfEMP1; *Plasmodium falciparum*

**Subject Categories** Microbiology, Virology & Host Pathogen Interaction

## Introduction

Despite the significant reductions in mortality and morbidity of malaria in the last decade, the percentage of patients infected with *Plasmodium falciparum* that succumb to severe malaria (SM) is not changing (WHO, 2017), with cerebral malaria (CM) contributing to much of the mortality. The overall mortality rate for CM in children is 15–25%, with a recent MRI study showing that brain swelling is strongly associated with fatal outcome in CM (Seydel *et al*, 2015). The pathology of CM has been studied extensively (Idro *et al*, 2005; Hawkes *et al*, 2013) but also debated for many decades, as discussed in numerous reviews (Shikani *et al*, 2012; Cunnington *et al*, 2013; Storm & Craig, 2014; Wassmer & Grau, 2017). What is clear is that the pathogenesis is multifactorial, with a role for the immune response to the *Plasmodium* infection (Hunt & Grau, 2003; Ioannidis *et al*, 2014; Dieye *et al*, 2016; Mandala *et al*, 2017; Wolf *et al*, 2017) and obstruction of the microvasculature by sequestration and rosetting (Rowe *et al*, 2009; Craig *et al*, 2012; Ponsford *et al*, 2012; White *et al*, 2013; Milner *et al*, 2015), leading to endothelial dysfunction. Sequestration of *P. falciparum*-infected erythrocytes (IE) in brain microvasculature is a hallmark of human CM as shown in post-mortem studies (Pongponratn *et al*, 1991; Taylor *et al*, 2004), but whether this sequestration is due to differential binding of IE to brain endothelium has been harder to demonstrate.

The major mediator of parasite cytoadherence to endothelium is *P. falciparum* erythrocyte membrane protein 1 (PfEMP1), a variant surface antigen expressed on knobs on the IE surface and encoded by approximately 60 *var* genes per parasite genome, with only one PfEMP1 being expressed on the surface of any individual IE (Scherf *et al*, 2008; Pasternak & Dzikowski, 2009). PfEMP1 is composed of multiple Duffy binding-like (DBL) and cysteine-rich interdomain region (CIDR) domains and can be classified into four main groups A, B, C and E based on the 5′ upstream sequence of the encoding *var* gene (Fig 1; Smith, 2014). PfEMP1 binds to a range of receptors and includes the mutually exclusive CD36 and endothelial protein C

1   Department of Parasitology, Liverpool School of Tropical Medicine, Liverpool, UK
2   Malawi-Liverpool-Wellcome Trust Clinical Research Programme, Blantyre, Malawi
3   College of Medicine, University of Malawi, Blantyre, Malawi
4   Department of International Health, Immunology & Microbiology, Centre for Medical Parasitology, University of Copenhagen, Copenhagen, Denmark
5   Department of Infectious Diseases, Rigshospitalet, Copenhagen, Denmark
6   Blantyre Malaria Project, College of Medicine, University of Malawi, Blantyre, Malawi
7   Department of Osteopathic Medical Specialties, College of Osteopathic Medicine, Michigan State University, East Lansing, MI, USA
8   Institute of Infection and Global Health, University of Liverpool, Liverpool, UK
9   Wellcome Centre for Molecular Parasitology, Institute of Infection, Immunity and Inflammation, College of Medical Veterinary & Life Sciences, University of Glasgow, Glasgow, UK
    *Corresponding author. Tel: +44 151 705 3297; E-mail: janet.storm@lstmed.ac.uk
    **Corresponding author. Tel: +44 151 705 3161; E-mail: Alister.Craig@lstmed.ac.uk

receptor (EPCR)-binding phenotypes, mediated by N-terminal CIDR domains (Kraemer & Smith, 2006; Semblat et al, 2006, 2015; Rask et al, 2010; Hviid & Jensen, 2015). Approximately half of group A PfEMP1 and a subset of group B/A chimeric PfEMP1, also known as domain cassette 8 (DC8), bind to EPCR via CIDRα1 domains, whereas group B and C PfEMP1 bind CD36 via CIDRα2-CIDRα6 domains. In addition, binding to intercellular adhesion molecule 1 (ICAM-1) is mediated via DBLβ domains adjacent to the CIDR domains and in some cases has been associated with a dual-binding phenotype with EPCR (Lennartz et al, 2017).

In choosing which host receptors to study, we took into account the findings that categories of PfEMP1 types are also associated with *in vivo* expression in SM. A particularly strong example of this is where parasites expressing *var* genes encoding PfEMP1 containing EPCR-binding domains have shown a strong association with the development of SM, including CM (Avril et al, 2012; Claessens et al, 2012; Lavstsen et al, 2012; Bengtsson et al, 2013; Bertin et al, 2013; Jespersen et al, 2016; Kessler et al, 2017; Mkumbaye et al, 2017). *In vitro*, parasites expressing EPCR-binding PfEMP1 show greater degrees of binding to EPCR, as well as to ICAM-1 receptors, both of which are expressed on brain microvascular endothelium (Turner et al, 2013; Avril et al, 2016; Lennartz et al, 2017). ICAM-1 binding has been mapped to some, but not all, DBLβ domains found

adjacent to the N-terminal CIDR domains in about one-third of all PfEMP1. A subset of ICAM-1-binding DBLβ domains were recently shown to be specific for group A EPCR-binding PfEMP1 and found to be expressed at higher levels in parasites from CM patients than in parasites from non-CM patients (Lennartz et al, 2017). Parasites expressing CD36-binding PfEMP1 are found in many patient isolates regardless of symptoms, although some data suggest that they may constitute a smaller proportion of parasites in SM patients (Heddini et al, 2001; Ndam et al, 2017), and are not seen in parasite isolates taken from women with placental malaria (Smith et al, 2013). Several other host receptors for PfEMP1 have been described, however, while PfEMP1 proteins that bind these receptors have been identified (Berger et al, 2013), links between cytoadherence and paediatric CM have not been established, and these were not tested in our study.

Infected erythrocytes binding to specific receptors during an infection may have different functional consequences on the endothelium and hence on disease severity. One clear example is that by binding to EPCR, the IE interfere with production of activated protein C thereby launching the coagulation cascade, leading to increased thrombin production (Moxon et al, 2013) and the potential to cause pro-inflammatory PAR1-mediated endothelial activation (Petersen et al, 2015; Gillrie et al, 2016). Other

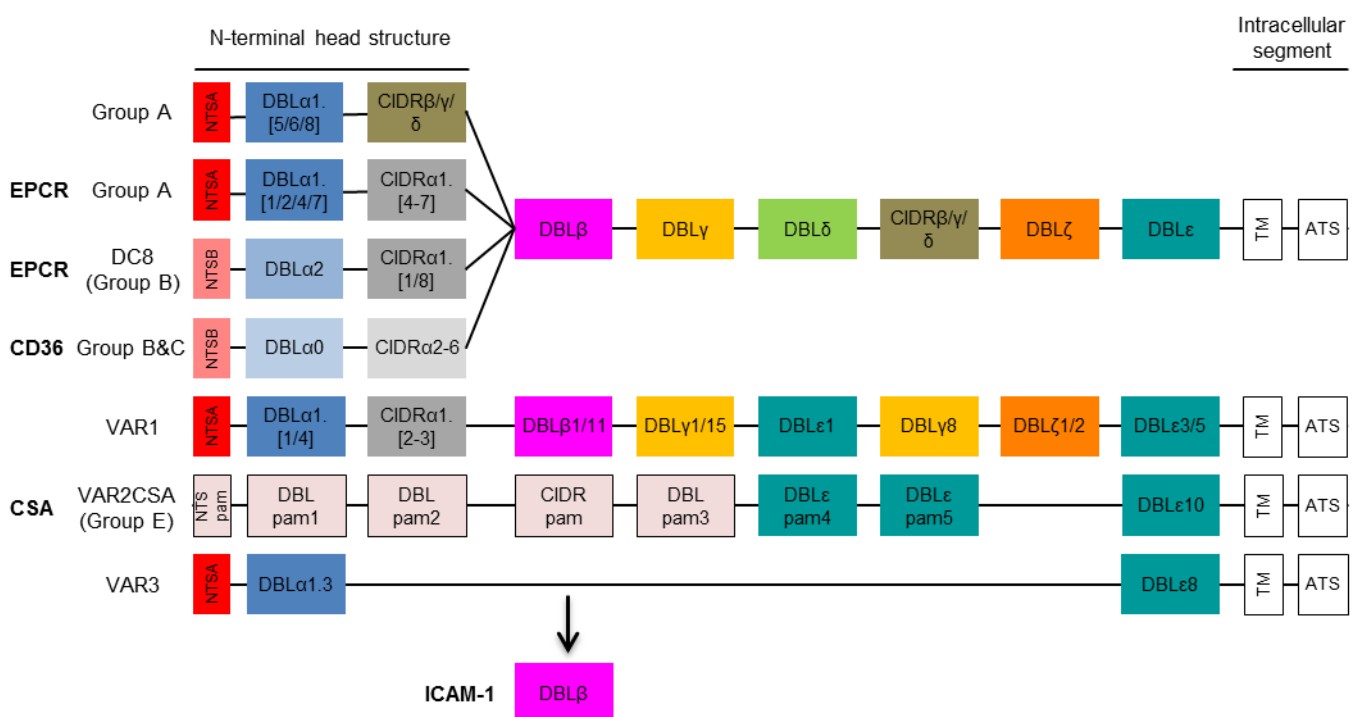

**Figure 1. PfEMP1 domain structure.**

A schematic presentation of PfEMP1 domain structure comprising a N-terminal head structure, 2-6 subsequent C-terminal domains, a transmembrane domain (TM) and an intracellular acidic terminal segment (ATS) with known receptors indicated in bold. Receptor specificity is determined by combined DBL and CIDR domains with mutually exclusive binding to EPCR and CD36 by different CIDRα domains in the head structure. Part of group A PfEMP1 and a particular subset of group B/A chimeric PfEMP1 (DC8) bind to EPCR via CIDRα1 domains, whereas group B and C PfEMP1 bind CD36 via CIDRα2-6 domains. The atypical group E VAR2CSA PfEMP1 binds placental chondroitin sulphate A (CSA) via DBLpam1 and DBLpam2 domains. The binding phenotype of VAR 1, VAR3 and group A CIDRβ/γ/δ domains is unknown, but they do not bind EPCR or CD36. DBLβ domains can be involved in ICAM-1 binding and are from both groups A and B. Not much is known about the other DBL domains (γ/δ/ε/ζ), but the DBLε and DBLζ domains are implicated in IgM and $\alpha_2$-macroglobulin binding.

PfEMP1-receptor interactions have been shown to activate signalling pathways in endothelial cells (Wu *et al*, 2011; Gillrie & Ho, 2017), but the effect of these events on pathology is unclear. More recent work has also suggested that as well as cytoadherence-mediated events, the accumulation of sequestered IE in vessels may facilitate endothelial dysfunction caused by the local release of soluble mediators following schizont rupture (Gallego-Delgado & Rodriguez, 2017).

Thus, it remains unclear why a particular child develops CM at a particular time, as the vast majority of *P. falciparum* infections do not lead to CM. In addition, most African children who develop CM have had malaria previously without developing CM. One possible mechanism to explain why a child develops CM at a particular time is that they have been infected with a *P. falciparum* variant that facilitates recruitment of IE to endothelium in the brain. While multiple lines of evidence indicate that specific PfEMP1 variants are associated with severe malaria, that association has not been substantiated by directly measuring the binding of IE to endothelial cells (EC). Thus, the question as to whether IE from children with CM have cytoadherence properties that enable them to bind to brain endothelium and thus enhancing their sequestration in that site has not been tested. The extent of sequestration in the brain is unknown for non-CM cases, although post-mortem observations of brain vessels from malaria-infected children dying from other causes of coma (not CM) show much lower levels of IE sequestration than CM cases (Milner *et al*, 2015). Therefore, as a comparison, isolates from children with uncomplicated malaria (UM) have also been tested for their binding phenotype in the present study.

A number of studies have investigated cytoadherence of specific PfEMP1 variants to human microvascular endothelial cells, but these have been with laboratory strains or PfEMP1-modified parasites (Madkhali *et al*, 2014; Gillrie *et al*, 2015). Patient isolates have also been investigated for their binding phenotype, but mainly on purified protein and mostly under static conditions (Craig *et al*, 2012; Almelli *et al*, 2014; Mahamar *et al*, 2017; Ndam *et al*, 2017). While providing important evidence, these studies were unable, largely for technical reasons, to combine the most appropriate target (primary brain endothelium) and parasite isolates as close to the patient sample as possible, with a physiologically relevant assay. To address our hypothesis that CM is driven by increased binding of IE to brain endothelium, we assessed whether IE freshly isolated from circulating blood of children with CM preferentially bound TNF-activated primary brain microvascular endothelium, compared to IE isolated from UM children. We postulated that such a difference might be associated with the expression of particular PfEMP1 variants and with binding to specific endothelial receptors. We collected IE from carefully characterised paediatric CM and UM cases in Malawi and determined cytoadherence to primary human microvascular endothelial cells, with minimal *in vitro* expansion of the parasite population, using a microfluidic flow device, an experimental design reflecting *in vivo* physiology. Expression of PfEMP1 variants was investigated by qPCR using the most up-to-date set of *var* domain type-specific primers available to us.

To our knowledge, this study is the first study to employ such a comprehensive approach to address the question of whether cytoadherence is involved in the pathogenesis of CM.

# Results

## Recruitment of study participants

Children were recruited over three malaria seasons from 2013 to 2015 using the selection criteria described in the Materials and Methods section. Total CM cases admitted to the research ward have been decreasing since 2010, from 165 cases to 48 (18) cases in 2013, 78 (26) in 2014 and 43 (14) in 2015. Numbers in brackets are the recruited number of children for our cytoadherence study. To improve the specificity of the clinical diagnosis of CM, only children with at least one feature of malarial retinopathy (Maccormick *et al*, 2014) were included, resulting in the recruitment of a total of 58 cases. A total of 53 UM cases, matched on an annual basis to the number of CM cases, were included. Clinical characteristics of the total UM and CM cohorts and the cases used for experiments are summarised in Table 1. The median age of children with UM was higher than children with CM. Compared to children with UM, children with CM had significantly higher median pulse and respiratory rates, higher median lactate concentration and lower median haematocrit levels, indicators of severe disease (WHO, 2016). Ten of the children with CM (17%) died. To achieve 2% parasitaemia needed for the cytoadherence assays, only blood samples from children with at least 2% peripheral parasitaemia were utilised. The clinical characteristics of these selected cases were similar to the overall cohort of children with each of these clinical syndromes.

## Cytoadherence of clinical isolates to microvascular endothelial cells under flow

Isolated IE were cultured until the parasites were at the trophozoite stage, when PfEMP1 is expressed on the surface of the IE, and a suspension of 2% parasitaemia and 5% haematocrit was prepared. Using the microfluidic device, cytoadherence to primary human microvascular endothelial cells, derived from brain (HBMEC) and dermis (HDMEC), was determined under flow conditions. Isolates from CM cases demonstrated an average binding of 110 IE/mm$^2$ (95% CI: 37–182) to HBMEC which was significantly higher ($P = 0.041$) than HBMEC binding of UM cases at 43 IE/mm$^2$ (95% CI: 28–57; Fig 2). In contrast, there was no difference in binding to HDMEC ($P = 0.171$) between IE from CM cases (average 165 IE/mm$^2$, 95% CI: 81–250) and UM cases (average 110 IE/mm$^2$, 95% CI: 71–149). Binding of UM isolates to HBMEC was significantly lower compared to HDMEC ($P = 0.002$), which was not the case for CM isolates. For isolates from CM patients, avid binding was a common feature; isolates that bound well to HBMEC also bound well to HDMEC with a Spearman's correlation coefficient of 0.83 ($P < 0.0001$). For UM isolates, however, there was no correlation between binding to HBMEC and HDMEC ($r = 0.20$, $P = 0.28$). A recent publication by Azasi *et al* (2018b) showed that DC8-PfEMP1 expressing IE do not bind EPCR in the presence of normal human serum. Therefore, we tested whether adding 10% human serum to the binding buffer would decrease the cytoadherence of selected patient isolates to HBMEC (Appendix Fig S2). Human serum did not change the binding of three patient isolates that showed significant EPCR binding to HBMEC. The binding of DC8 variant IT4var19 was also not affected by the addition of human serum in our flow assay system.

**Table 1.    Clinical characteristics of the study participants.**

| | Total cohort uncomplicated malaria (*n* = 53) | Total cohort cerebral malaria (*n* = 60) | *P*-value | Used in assay uncomplicated malaria (*n* = 35) | Used in assay cerebral malaria (*n* = 27) | *P*-value |
|---|---|---|---|---|---|---|
| Age, months | 51 (30–74) | 42 (24–59) | 0.04 | 53 (38–89) | 36 (23–50) | 0.005 |
| Gender, % female | 53 | 47 | | 51 | 48 | |
| Axillary temperature, °C | 38.8 (38.2–39.4) | 39.0 (38.1–40.0) | | 38.8 (381–39.4) | 39.0 (37.9–40.0) | |
| Pulse rate, beats/min | 124 (107–140) | 157 (140–175) | < 0.0001 | 124 (114–146) | 156 (133–175) | 0.0001 |
| Systolic blood pressure, mmHg | 98 (91–102) | 95 (86–104)[a] | | 97 (91–99) | 94 (85–103)[b] | |
| Respiratory rate, breaths/min | 30 (25–30) | 40 (36–52) | < 0.0001 | 28 (24–30) | 41 (32–52) | < 0.0001 |
| Blood glucose, mmol/l | 5.7 (5.1–6.4) | 5.4 (4.4–6.8) | | 5.8 (5.1–6.5) | 5.0 (4.5–6.7) | |
| Blood lactate, mmol/l | 2.9 (2.0–3.4) | 4.5 (2.4–8.8) | < 0.0001 | 2.9 (1.9–3.5) | 4.6 (2.3–8.2) | 0.0006 |
| Haematocrit, % | 35.0 (30.0–38.0) | 22.0 (18.0–25.6) | < 0.0001 | 36.0 (29.5–38.0) | 20.4 (16.8–25.1) | < 0.0001 |
| HRP2, μg/ml | | 7.1 (2.2–9.9)[c] | | | 9.5 (3.1–11.0)[d] | |
| Parasitaemia, parasites/μl (×10⁴) | | 11.8 (4.2–38.4) | | | 29.8 (14.6–59.9) | |
| Platelets/μl (×10⁴) | | 4.9 (2.4–9.6)[c] | | | 4.9 (2.4–9.8)[d] | |

Shown are the median with the interquartile range in brackets for the total cohort and the cases used in the binding assays. For each variable, statistical differences between UM and CM cases were determined by Mann–Whitney *U*-test (continuous variables) or Fisher's exact test (categorical variables), and *P*-values < 0.05 are indicated. HRP2 = histidine-rich protein 2. Group size in a = 50, b = 21, c = 55 and d = 24 children.

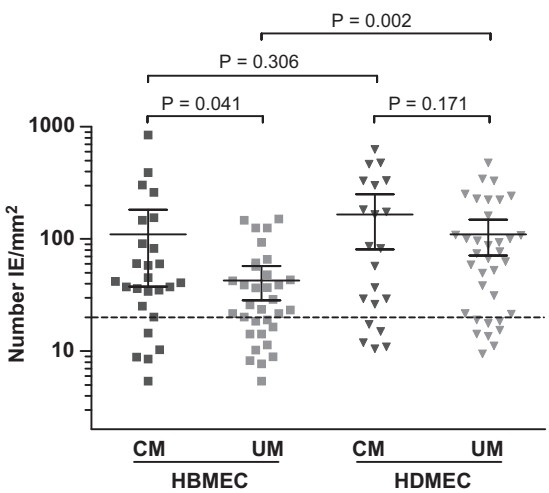

**Figure 2.   Cytoadherence of IE from CM and UM cases to HBMEC and HDMEC.**

IE were isolated, and binding to HBMEC and HDMEC was determined under flow conditions. Number of IE bound per mm² EC surface was calculated and shown is the mean ± 95% CI for 26 CM and 33 UM cases on HBMEC and 21 CM and 35 UM cases on HDMEC on a log scale. *P*-value was calculated by two-tailed unpaired *t*-test. The dotted line is 20 IE/mm², the cut-off value for inclusion of the inhibition data.

### Inhibition of cytoadherence to microvascular endothelial cells under flow

To assess the differential role of the endothelial receptors ICAM-1, EPCR and CD36, binding was determined in the presence of inhibitory antibodies, αICAM-1 and αCD36, or recombinant protein, rEPCR (Fig 3). Paired analysis of the inhibition binding data is shown in Fig 3A and significant inhibition was observed for all the

EC-inhibitor combinations, except for inhibition of binding of UM isolates to HDMEC by rEPCR. The data are summarised as percentage inhibition in Fig 3B–D, to compare receptor-dependent adherence between CM and UM. Approximately half of the IE displayed ICAM-1-dependent binding (> 50% inhibition) to both HBMEC and HDMEC, but there was no significant difference between CM and UM isolates nor between the dependency of ICAM-1 binding to HBMEC and HDMEC. CD36 expression is extremely low on primary HBMEC (Avril *et al*, 2016), so studies on binding to CD36 were only performed with HDMEC. Inhibition of cytoadherence by αCD36 antibody was variable and not significantly different between the CM and UM isolates (*P* = 0.23), although there was a trend for higher CD36-dependent binding in UM isolates. EPCR-dependent binding also varied, with a subset of isolates binding particularly well to EPCR. This was more pronounced for CM isolates binding to HBMEC, but not significantly different from UM isolates (*P* = 0.073). There was also no significant difference between rEPCR inhibition of binding to HBMEC and HDMEC. For a few isolates, there was more binding in the presence of αICAM-1 antibody or rEPCR compared to binding in the absence of inhibitor (Fig 3B and D), and this was more often the case for UM isolates. The reason for this is unclear; we were unable to collect the bound IE to investigate this phenomenon further.

### Correlations between IE binding and clinical parameters

We assessed the association between the *in vitro* cytoadherence properties of the IE and clinical parameters associated with sequestration. For the CM cases, the degree of IE binding to HDMEC was positively correlated with peripheral parasite density at recruitment (*r* = 0.56, *P* = 0.011). Binding of IE to HBMEC was less clearly associated with peripheral parasite density (*r* = 0.40, *P* = 0.056). Differences in levels of binding were not due to variation in parasitaemia of the cultured IE, as the binding assay was performed at a standardised 2% parasitaemia. Binding of CM isolates was

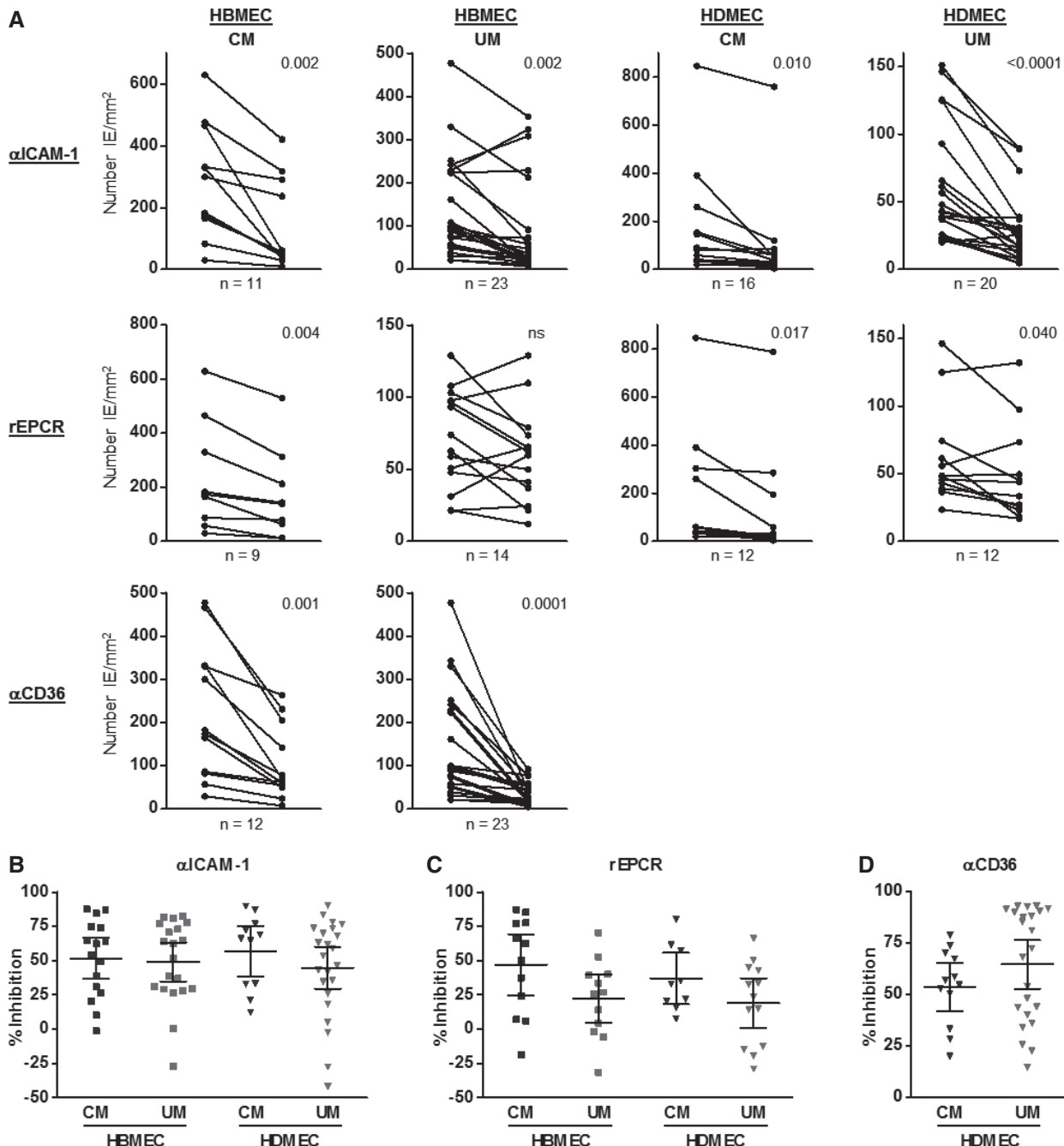

**Figure 3.   Inhibition of cytoadherence of IE from CM and UM cases to HDMEC and HBMEC by αICAM-1 and αCD36 antibody and rEPCR.**

A    IE were isolated, and binding to HBMEC and HDMEC was determined under flow conditions in the absence and presence of 5 µg/ml αICAM-1 or αCD36 antibody or 50 µg/ml rEPCR. Number of IE bound per mm² EC surface was determined.

B    Using the same data, percentage inhibition by αICAM-1 antibody was calculated relatively to binding in the absence of antibody.

C    Using the same data, percentage inhibition by rEPCR was calculated relatively to binding in the absence of inhibitor.

D    Using the same data, percentage inhibition by αCD36 antibody was calculated relatively to binding in the absence of antibody.

Data information: (A) shown is the paired analysis between the absence and presence of inhibitor, with number of cases (*n*) indicated. Statistical significance was determined by two-tailed paired *t*-test, and the *P*-value is shown; ns is not significant. (B–D) shown are the mean ± 95% CI, and no significant differences were determined with a two-tailed unpaired *t*-test. Each assay was only conducted once for each isolate. The number of isolates tested can be seen from the dot plot.

negatively correlated with peripheral platelet levels for both binding to HDMEC ($r = -0.66$, $P = 0.001$) and HBMEC ($r = -0.56$, $P = 0.005$). We were unable to assess the association between IE adhesion and fatal outcome as only three children for whom we had binding data died. None of the other clinical characteristics, including histidine-rich protein 2 concentrations, showed any significant

correlation with the cytoadherence of IE. For the UM cases, none of the parameters assessed were significantly correlated with cytoadherence. Peripheral parasite density (as per WHO standard) and platelet counts were not determined at admission in UM cases; however, after processing the UM blood samples, parasitaemia was determined by microscopy of Giemsa-stained smears and no correlation was found between parasitaemia and binding intensity.

### Analysis of *var* gene transcripts

To analyse the PfEMP1 domain structure (Fig 1) of the patient isolates, transcript levels of the coding *var* genes were determined. qPCR was performed and *var* transcript values (Tu) were calculated and compared between CM and UM cases (Table 2). A detailed description of coverage, sensitivity and limitations of the primer set is published in Mkumbaye *et al* (2017) and summarised in Appendix Table S1. There were significantly higher levels of transcripts encoding CIDRα1 domains in CM compared to UM cases (median Tu = 104.6 for CM versus 9.3 for UM, $P < 0.001$). This was also true when comparing levels of CIDRα1-encoding group B/A (DC8, CIDRα1.DC8: median Tu = 33.9 for CM versus 4.6 for UM, $P < 0.001$) or group A transcripts (CIDRα1.A: Tu = 43.6 for CM and 4.3 for UM, $P < 0.001$). Transcript levels of some of the individual CIDRα1 domain types, such as CIDRα1.1, CIDRα1.4, CIDRα1.5a, CIDRα1.6a and CIDRα1.7, were also significantly higher in CM cases. Primers targeting the non-EPCR-binding group A N-terminal CIDRδ reported higher transcript levels in CM, whereas primers for group A CIDRγ or CD36-binding group B and C CIDR domains did not. The two primer sets DBLα2/1.1/1.2/1.4/1.7 and DBLα1.5/1.6/1.8, predicted to mainly target CIDRα1 or CIDRβ/γ/δ-containing PfEMP1, respectively, reported higher levels in CM compared to UM, with relative levels of DBLα2/1.1/1.2/1.4/1.7 markedly higher than DBLα1.5/1.6/1.8 levels. Data from primers targeting C-terminal DBL domains showed that, although median transcript levels were relatively low, *var* genes encoding ICAM-1-binding domains of both group A (DBLβ1/3-1) and group B (DBLβ5) PfEMP1 were found at significantly higher levels in CM versus UM cases, as were transcripts encoding group A DC5, which has been linked to PECAM-1 binding (Berger *et al*, 2013). Likewise, the transcript levels of some of the DBLε and DBLζ domains: DBLζ2a, DBLζ2c, DBLζ3, DBLζ5 and DBLε2, were low, but significantly higher in CM cases. These domains have been implicated in non-immune IgM and $\alpha_2$-macroglobulin binding (Jeppesen *et al*, 2015; Stevenson *et al*, 2015a,b; Pleass *et al*, 2016).

### Correlations between IE binding and *var* gene expression data

We next investigated correlations between adhesion phenotypes and *var* type expression data. Correlation coefficients rho (r) were calculated for the Tu values of each primer cocktail with the binding data, separated in binding to HBMEC or HDMEC of CM or UM isolates. *r* was calculated by the Spearman correlation test, but with relatively small group sizes not many *P*-values were < 0.05, and after correction for multiple comparisons none of the correlations was statistically significant. Appendix Table S2 shows the *r*-values for the 15 correlations that were significant prior to correcting for multiple comparisons, which were mainly detected for CM isolates and for receptor-dependent binding. We recognise the lack of power

of this study to identify significant associations but provide these data to guide future research focussed on the role of the domains, including those with less well-known functions, such as the DBLζ and DBLε domains.

## Discussion

Post-mortem studies have shown that IE sequestration in the brain is a feature of CM (Taylor *et al*, 2004; Ponsford *et al*, 2012) and have shown a strong association between IE sequestration and key pathogenic processes occurring in CM: endothelial activation/inflammation (Turner *et al*, 1994), microvascular thrombosis (Dorovini-Zis *et al*, 2011; Moxon *et al*, 2013, 2015) and endothelial barrier disruption (Dorovini-Zis *et al*, 2011). Children who died of CM also have sequestered IE in the retinal microvessels, which correlates with the amount of sequestration in the brain (Barrera *et al*, 2015). To assess the role of cytoadherence in CM pathology, some studies have found correlations between the *in vitro* adherent properties of patient isolates with specific host receptors (Newbold *et al*, 1997; Rogerson *et al*, 1999; Heddini *et al*, 2001; Mayor *et al*, 2011; Ochola *et al*, 2011; Almelli *et al*, 2014), and the importance of specific PfEMP1 variants has been identified using parasite isolates from patients with different disease syndromes. Based on the hypothesis of cytoadherence to brain endothelium being involved in CM pathogenesis and following a process involving selecting and panning IE on HBMEC (Claessens *et al*, 2012; Lavstsen *et al*, 2012; Turner *et al*, 2013), EPCR was identified as the main receptor of PfEMP1 DC8 and DC13 variants responsible for HBMEC binding. It is worth noting that despite extensive evidence of the involvement of these PfEMP1 variants in SM by qPCR genotyping, a recent publication has questioned whether EPCR acts as a primary adhesion receptor to endothelium for some of these PfEMP1 variants (Azasi *et al*, 2018b). Taken together, these data, showing that severe malaria is associated with parasite variants with particular cytoadherence characteristics, provide strong evidence that sequestration may be a critical process in the pathogenesis of CM.

To look more directly at the role of cytoadherence in disease, we assessed whether parasite isolates from well-defined clinical cases of CM, refined by observation of malaria retinopathy (Maccormick *et al*, 2014), bind human brain microvascular endothelium under physiological flow conditions more readily than do those from UM controls. Our main finding is that parasite isolates from CM and UM patients exhibit differential binding capacities to primary brain endothelial cells, in particular that UM isolates bind less well to HBMEC than CM isolates. For CM patient isolates, binding intensity also correlated with peripheral parasitaemia and the degree of thrombocytopenia. We examined whether this cytoadherence phenotype is determined by the expression of specific PfEMP1 variants by performing *var* gene analysis by qPCR. Higher transcript levels of ICAM-1- and EPCR-binding *var* gene domains were detected in CM isolates, which for the EPCR-binding *var* domains seems to correlate with EPCR-dependent binding to HBMEC. We also observed potential associations between cytoadherence and the transcript levels of the DBLζ and DBLε domains, indicating new roles for these domains in receptor-mediated cytoadherence, but focussed studies on these are needed to confirm our initial observations.

**Table 2.  Transcript levels of *var* gene domains by patient group.**

| Target | | | UM | | CM | | |
|---|---|---|---|---|---|---|---|
| Domain | Group | Predicted receptor | *n* | Tu | *n* | Tu | *P*-value |
| CIDRα1.1 | B/A | EPCR | 34 | 2.6 (1.0–28.9) | 22 | 24.1 (2.0–114.9) | < 0.0001 |
| CIDRα1.8a | B/A | EPCR | 30 | 1.0 (1.0–24.4) | 23 | 2.9 (1.0–75.0) | 0.040 |
| CIDRα1.8b | B/A | EPCR | 31 | 1.0 (1.0–8.5) | 23 | 1.0 (1.0–16.2) | |
| **Sum: CIDRα1.DC8** | B/A | EPCR | 31 | 4.6 (1.0–37.1) | 22 | 33.9 (4.7–166.1) | < 0.0001 |
| CIDRα1.4 | A | EPCR | 34 | 1.0 (1.0–5.6) | 24 | 3.0 (1.0–29.7) | 0.002 |
| CIDRα1.5a | A | EPCR | 33 | 1.0 (1.0–2.1) | 23 | 1.0 (1.0–30.1) | 0.004 |
| CIDRα1.5b | A | EPCR | 34 | 1.0 (1.0–15.5) | 23 | 4.6 (1.0–105.0) | |
| CIDRα1.6a | A | EPCR | 31 | 1.0 (1.0–1.0) | 23 | 1.0 (1.0–20.2) | 0.005 |
| CIDRα1.6b | A | EPCR | 31 | 1.0 (1.0–8.5) | 23 | 1.0 (1.0–8.4) | |
| CIDRα1.7 | A | EPCR | 34 | 1.2 (1.0–10.4) | 23 | 8.5 (1.0–35.4) | 0.0007 |
| **Sum: CIDRα1.A** | A | EPCR | 30 | 4.3 (1.0–35.6) | 23 | 43.6 (6.9–136.7) | < 0.0001 |
| **Sum:CIDRα1_all** | A & B/A | EPCR | 30 | 9.3 (1.0–81.6) | 23 | 104.6 (14.2–218.2) | < 0.0001 |
| DBLζ2a | | | 31 | 1.0 (1.0–3.1) | 23 | 1.3 (1.0–6.6) | 0.043 |
| DBLζ2b | | | 25 | 1.0 (1.0–2.8) | 19 | 1.0 (1.0–6.6) | |
| DBLζ2c | | | 25 | 1.0 (1.0–3.9) | 19 | 1.0 (1.0–32.4) | 0.048 |
| DBLζ3 | | | 31 | 1.5 (1.0–9.4) | 23 | 3.2 (1.1–24.0) | 0.009 |
| DBLζ4 | | | 31 | 4.3 (1.0–75.2) | 23 | 6.3 (1.1–55.0) | |
| DBLζ5 | | | 31 | 1.4 (1.0–17.9) | 23 | 6.6 (1.0–129.9) | 0.011 |
| DBLζ6 | | | 25 | 1.5 (1.0–10.0) | 18 | 3.8 (1.0–27.0) | |
| **Sum: DBLζ_all** | | | 25 | 21.3 (3.4–95.3) | 19 | 44.0 (13.9–389.1) | 0.035 |
| DBLε2 | | | 25 | 1.0 (1.0–1.0) | 19 | 1.0 (1.0–5.4) | 0.004 |
| DBLε6 | | | 25 | 1.4 (1.0–9.8) | 19 | 5.7 (1.0–33.2) | |
| DBLε11 | | | 25 | 1.0 (1.0–15.3) | 19 | 1.3 (1.0–23.7) | |
| DBLε13 | | | 31 | 8.9 (1.0–87.6) | 23 | 14.2 (1.1–126.8) | |
| DBLε14 | | | 25 | 1.0 (1.0–1.0) | 19 | 1.0 (1.0–2.9) | |
| **Sum: DBLε_all** | | | 25 | 16.8 (1.06–123.8) | 19 | 36.1 (4.1–147.8) | |
| DBLβ1/3-1 | A | ICAM-1 | 33 | 1.0 (1.0–7.7) | 23 | 6.6 (1.0–24.1) | 0.0001 |
| DBLβ1/3-2 | A | (ICAM-1) | 34 | 1.3 (1.0–30.8) | 23 | 1.5 (1.0–29.1) | |
| DBLβ5 | B | ICAM-1 | 34 | 2.1 (1.0–26.1) | 23 | 4.7 (1.0–31.2) | 0.021 |
| DC5 | A | PECAM-1 | 32 | 1.0 (1.0–19.3) | 23 | 5.3 (1.0–61.9) | 0.024 |
| CIDRα3.1/3.2 | B/C | CD36 | 34 | 1.1 (1.0–7.8) | 23 | 2.0 (1.0–8.0) | |
| CIDRδ | A | | 33 | 2.4 (1.0–25.0) | 23 | 13.7 (1.4–69.7) | 0.003 |
| CIDRγ3.1 | A | | 28 | 1.0 (1.0–6.5) | 19 | 1.0 (1.0–1.5) | |
| DBLα1all | A | | 34 | 31.1 (3.2–111.3) | 23 | 90.5 (35.2–257.8) | < 0.0001 |
| DBLα2/1.1/1.2/1.4/1.7 | A | | 34 | 32.3 (6.4–117.7) | 23 | 134.8 (26.7–428.1) | < 0.0001 |
| DBLα1.5/1.6/1.8 | A | Non-EPCR | 34 | 16.5 (2.3–66.9) | 23 | 46.9 (4.9–107.6) | 0.0006 |
| var2csa | E | CSA | 25 | 4.8 (1.0–44.7) | 19 | 8.7 (1.0–135.3) | |
| var3 | A | | 25 | 1.0 (1.0–5.2) | 19 | 1.0 (1.0–6.8) | |
| CIDRα1.2-K | | | 23 | 1.0 (1.0–1.4) | 18 | 1.0 (1.0–1.4) | |
| CIDRα1.2-K+CIDRα1.3-K | | | 31 | 1.0 (1.0–3.1) | 23 | 1.0 (1.0–4.0) | |

The transcript unit (Tu) was calculated for the primer sets and shown are the median, with the 10th and 90th percentiles in brackets. In bold are the specific groups, which are the sum of the Tu values of primers listed above it. For DBLζ_all, only the highest Tu value of either DBLζ4 or DBLζ6 was included. Receptor names in brackets are probable and left blank when unknown. Number of cases in the analysis (*n*) is specified, and statistical differences between UM and CM cases were determined by Mann–Whitney *U*-test and *P*-values < 0.05 are indicated.

Cerebral malaria isolates adhered equally well to HBMEC and HDMEC, and binding levels were highly correlated, in agreement with earlier work for laboratory strains that were selected for binding to HBMEC and which bound well to other types of non-brain microvascular EC (Claessens *et al*, 2012; Avril *et al*, 2013). This is also reflected by comparable levels of IE sequestration across many organs in the autopsy studies of children with CM (Seydel *et al*, 2006; Milner *et al*, 2014). The UM isolates bind well to HDMEC too, but less avidly to HBMEC, and there is no correlation between their binding to HBMEC and HDMEC. The overall significantly higher binding of CM isolates to HBMEC supports the role of intracerebral cytoadherence and sequestration of infected erythrocytes in CM. The reduced binding of UM isolates to HBMEC compared to binding to HDMEC would result in the majority of IE binding in locations other than the brain and thus avoiding CM pathology and high mortality risk. We hypothesise that UM isolates, which are thought to bind to CD36, do not adhere well to HBMEC because brain endothelial cells constitutively express little CD36 (Avril *et al*, 2016). The binding of patient isolates to purified receptors, including CD36, has been investigated in many studies, but mostly under static conditions, and the results are conflicting (Rogerson *et al*, 1999; Yipp *et al*, 2000; Heddini *et al*, 2001; Mayor *et al*, 2011; Ochola *et al*, 2011; Almelli *et al*, 2014). In our study, binding in the presence of αCD36 antibodies was only determined for HDMEC and there was no significant difference in CD36-dependent binding of CM or UM isolates. However, it is notable that there are two populations of UM isolates, with high and low CD36-dependent binding. There were no differences between the two patient populations in terms of clinical parameters or *var* gene transcripts with the available primers, but it would be interesting to analyse the *var* gene usage of the two sub-populations further.

Previous studies, including ours, suggested independent associations between binding to ICAM-1 (Ochola *et al*, 2011) and EPCR (Smith *et al*, 2013; Turner *et al*, 2013) with CM. Therefore, we used specific inhibitors of binding to these receptors to measure the dependency of binding via ICAM-1 and EPCR to endothelium in our assay system. There were no significant differences either in ICAM-1- or in EPCR-dependent binding between the UM and CM isolates, but there was a trend of higher EPCR-dependent binding of CM isolates to HBMEC ($P = 0.073$). The number of observations for binding inhibition by rEPCR was relatively low as we only had access to rEPCR from 2015. Paired analysis showed that there was no significant reduction in binding of UM isolates to HDMEC in the presence of rEPCR, indicating that EPCR does not play a major role in the adhesion of the UM isolates to HDMEC. EPCR binding by IE expressing the DC13 and DC8 variants has recently been questioned with data showing that DC13 expressing IE do not bind EPCR *in vitro* and binding of DC8 expressing IE to EPCR and brain endothelium was reduced in the presence of 10% human serum (Azasi *et al*, 2018). This effect of human serum was not seen with three patient isolates and It4var19 in our assay (Appendix Fig S2). This discrepancy may be explained by different assay conditions; Azasi *et al* determine static binding to the CD31-negative HBEC-5i cell line, while we used primary HBMEC under flow conditions.

This highlights a number of limitations in using *in vitro* methods to investigate the relationship between cytoadherence and disease. Besides cell and assay type, in standardising the *in vitro* assay parameters at 2% parasitaemia and 5% haematocrit under constant shear stress, we can only mimic *in vivo* dynamics, for which parasitaemia is variable and haematocrit is much higher at 30–50%. In our assay system, we could not collect the bound IE, hindering further molecular analysis of the adherent IE population. Refined flow adhesion techniques would be needed to overcome these limitations.

The IE that bind from the parasite suspension to HBMEC under flow conditions in our assays are a representation of the parasite types that might be recruited to brain endothelium from peripheral blood flow *in vivo*. In addition, parasite populations in peripheral blood, the only population readily accessible in patients, have been shown to resemble the same parasite populations as the sequestered IE (Montgomery *et al*, 2006). To ensure we were investigating brain-specific binding, we used TNF-activated primary HBMEC, with HDMEC as a comparator. That we did not find significant differences in receptor usage between the UM and CM isolates could be explained by the infection consisting of multiple *P. falciparum* genotypes, including those underlying *var* gene expression (Montgomery *et al*, 2006, 2007). It is also not as simple as a one-to-one interaction with one PfEMP1 variant binding to one specific receptor. Lately, dual receptor binding by one PfEMP1 variant has been reported, with some PfEMP1 being able to bind both EPCR and ICAM or CD36 and ICAM-1 simultaneously (Avril *et al*, 2016; Lennartz *et al*, 2017). We also determined binding of the patient isolates to HBMEC and HDMEC in the presence of αICAM-1 antibody and rEPCR combined, but the limited data (Appendix Fig S3) showed no significant difference between UM and CM isolates, nor between HBMEC and HDMEC binding. It is clear that HBMEC binding of some isolates is not affected by inhibition with αICAM-1 and rEPCR, indicating that other receptors may play a role in cytoadherence to brain endothelium, a phenomenon also seen with selected laboratory strains (Yipp *et al*, 2007; Avril *et al*, 2016; Mahamar *et al*, 2017; Metwally *et al*, 2017).

The domain structure of *var* genes expressed by the patient isolates was determined with a set of extensive qPCR primers. Transcripts encoding EPCR-binding domains, both when assessed by individual CIDRα1 domain subsets or summarised in the group B/A and A (CIDRα1.DC8, CIDRα1.A), as well as transcripts encoding the CIDRα1-associated DBLα domains, were all expressed higher in CM isolates. This was also the case for transcripts encoding group A non-EPCR-binding domains and the ICAM-1-binding domains DBLβ1/3-1 (group A) and DBLβ5 (group B), all in line with previous reports (Lavstsen *et al*, 2012; Bertin *et al*, 2013; Bernabeu *et al*, 2016; Kessler *et al*, 2017; Mkumbaye *et al*, 2017; Shabani *et al*, 2017). In our study, transcripts encoding other domains, found C-terminally in both group A and B PfEMP1, showed higher levels in CM compared to UM cases. This may be attributed to the inclusion of retinopathy in our CM cases, producing more definitive case definition. Two recent studies compared the *var* gene expression between retinopathy-positive and retinopathy-negative children and found a number of differences, concluding that group A *var* genes are more commonly found in the CM patient population that are retinopathy-positive (Abdi *et al*, 2015; Shabani *et al*, 2017). Higher transcripts of the DBLζ2a, DBLζ2c, DBLζ3, DBLζ5 and DBLε2 domains were detected in the CM cases, of which DBLζ2a, DBLζ3 and DBLε2 were also reported in a recent study (Kessler *et al*, 2017; Mkumbaye *et al*, 2017). The functions of the C-terminal DBLε and DBLζ domains are not fully understood, but some are involved in

non-immune IgM and $\alpha_2$-macroglobulin binding (Semblat *et al*, 2006, 2015; Jeppesen *et al*, 2015; Stevenson *et al*, 2015a; Pleass *et al*, 2016). The variant nature of *var* genes makes it difficult to design qPCR primers with high enough specificity and selectivity to cover transcripts encoding CD36-binding CIDR domains (group B and C *var* genes). For this reason and due to the different specificities and amplification bias of the primers, exact determination of *var* transcript distribution cannot be determined for each isolate (Mkumbaye *et al*, 2017).

We attempted to discover whether specific *var* domains contribute to the binding phenotype of the patient isolates by determining the correlation between IE binding and qPCR data. However, our study was not powered to examine these correlations with sufficient statistical significance after correction for multiple comparisons. The uncorrected results from this analysis, shown in Appendix Table S2, require further studies to identify statistically significant associations.

We identified an association between the expression of EPCR-binding PfEMP1 and IE cytoadherence to EPCR, but only for the CM isolates and mainly to HBMEC. While positive correlations provide straightforward evidence for associations, negative correlations can also be informative. The negative correlation between ICAM-1-dependent EC binding and levels of transcript encoding CIDRα1 domains (EPCR-binding) in UM isolates suggests that dual EPCR and ICAM-1 binding is not a prominent phenotype in UM. This indicates that EPCR is a major component of cytoadherence to HBMEC in CM, but not necessarily as the only primary receptor, as suggested by a recent study (Azasi *et al*, 2018). Recruitment of IE to the EC could also occur via other receptors and then subsequently bind to EPCR causing pathology linked to degradation of the control of coagulation pathways.

To validate the relevance of binding characteristics to pathogenesis, we assessed whether the level of cytoadherence *in vitro* corresponded with clinical characteristics in the patients from whom the parasites had been isolated. There was a correlation between the avidity of binding of CM isolates and peripheral parasitaemia and platelet counts at the time of admission of the CM cases. The increased binding (at standardised assay conditions of 2% parasitaemia and 5% haematocrit) to EC correlated with increased parasitaemia, but with decreased platelet numbers. The negative correlation between peripheral platelets and binding intensity is interesting as thrombocytopenia has been used as a predictor for malaria (Lampah *et al*, 2015; Thachil, 2017) and specifically for *P. falciparum* SM (Gerardin *et al*, 2002; Cserti-Gazdewich *et al*, 2012). A recent study in children recruited from the same hospital in 2015–2016 showed that low platelet levels were also associated with retinopathy-positive CM cases and brain swelling (Kessler *et al*, 2017). Some studies have shown that CM cases have more platelets localised in the brain microvasculature (Grau *et al*, 2003; Dorovini-Zis *et al*, 2011) and show increased platelet-mediated clumping of IE (Pain *et al*, 2001; Wassmer *et al*, 2008) and platelet involvement in the adhesion of IE to human microvascular EC lacking CD36 (Wassmer *et al*, 2004). We postulate that these processes lead to platelet sequestration and consumption, which is augmented by the presence of IE with high binding capabilities, and therefore a decrease in peripheral platelet counts. Activation of platelets and platelet consumption leads to a pro-coagulant state characterised by an increase in thrombin and von Willebrand factor, which are demonstrated in SM (Dorovini-Zis *et al*, 2011; O'Sullivan *et al*,

2016; Thachil, 2017). In addition, anti-coagulation and endothelial protective pathways are affected in CM through a decrease in EPCR and thrombomodulin and thus dysfunction of the activated protein C pathway (Moxon *et al*, 2013, 2015; O'Sullivan *et al*, 2016).

A number of mechanisms have been proposed to explain the pathology of CM, and it is unlikely that a single molecular process is involved. However, the identification of EPCR as a cytoadherence receptor and its association with CM in several studies, including this one, provides a compelling direction for future research.

Binding of IE to EPCR inhibits EPCR interaction with activated protein C, leading to deregulation of inflammation, coagulation and loss of endothelial barrier integrity (Gillrie *et al*, 2015; Sampath *et al*, 2015; Bernabeu *et al*, 2016; Kessler *et al*, 2017). This might be one of the reasons why the consequences of sequestration in the brain can be catastrophic. From our data and others, we do not find specific brain-binding IE in CM as CM isolates bind well to a variety of EC, which is also demonstrated by the sequestration of IE in many other organs (Avril *et al*, 2013; Milner *et al*, 2014, 2015). IE sequestration in the brain, irrespective of EPCR binding, leads to loss of EPCR function (Moxon *et al*, 2013); thus, the combined low levels of EPCR on HBMEC and binding of IE to EPCR lead to reduced levels of activated protein C and subsequent increase in thrombin, impacting on inflammation and coagulation, as reviewed in Bernabeu and Smith (2017). Furthermore, the endothelium responds locally to release of soluble content of sequestered IE leading to the disruption of barrier function (Gallego-Delgado & Rodriguez, 2017). Thus, cytoadherence of IE to brain endothelium may alter the balance of endothelial activation and protective pathways via a number of mechanisms, all driven by the binding properties of CM isolates to brain endothelium.

This work has shown that CM is associated with increased adhesion of IE to brain endothelium *in vitro* and that this adhesion is not specific to cerebral vascular endothelium. Indeed, it is the reduced ability of binding to brain EC that characterises isolates from UM patients that strengthens the case for the role of sequestration in the pathology of CM.

# Materials and Methods

### Recruitment of study participants

In this prospective study, children were recruited at the Queen Elizabeth Central Hospital, Blantyre, Malawi. The study was approved by the ethics committees of the College of Medicine, University of Malawi (protocol P.08/12/1264) and LSTM (protocol 12.29). Recruitment of CM cases took place at the Paediatric Research Ward under an overarching CM study. The inclusion criteria for this study were as follows: children aged between 1 and 12 years old; peripheral *P. falciparum* parasitaemia; at least 4+ on a thick blood smear (equals 10–100 parasites per high power field, equivalent of 1–10% parasitaemia. 1% parasitaemia is equivalent to 100,000 parasites/ μl); Blantyre Coma Score ≤ 2 on admission and no other causes of coma; at least one specific feature of malarial retinopathy, as determined by funduscopic examination by a trained clinician (White *et al*, 2001; Maccormick *et al*, 2014); and informed consent from the accompanying parent or guardian. Recruitment of UM cases took place at the paediatric A&E department, and inclusion criteria

were as follows: age between 1 and 12 years old, peripheral *P. falciparum* parasitaemia of at least 4+, fever, Blantyre Coma Score of 5 and informed consent from the parent or guardian. Exclusion criteria for both groups were as follows: gross signs of malnutrition, clinical manifestations of advanced HIV/AIDS and for UM and any features of severe malaria, as defined by WHO (2000). The study was in compliance with the principals of the Declaration of Helsinki and the Belmont Report. The following clinical variables were documented for each anonymised patient: age, sex, weight, temperature, respiratory and pulse rate, blood pressure, blood glucose and lactate concentrations, packed cell volume, duration of fever, medications in the last 2 weeks and hospital admissions in past year; and additionally for CM: HIV status, PfHRP2, platelets per μl blood and parasite density per μl blood (calculated from thick blood smears according WHO standards). Venous blood, ≤ 1 ml for CM and 3–4 ml for UM, was collected in citrate tubes and stored at 4°C until processing. Treatment of the children was according to national guidelines with artesunate replacing quinine in 2014 for CM cases and in 2015 for UM cases. As a consequence, most cases admitted to the research ward in 2015 had received treatment prior to study recruitment, thus affecting parasite viability. Therefore, a small number of cases from 2008 were shipped to LSTM, of which data for two isolates were obtained.

### Processing of whole blood

Blood was centrifuged at 500 *g* for 5 min, plasma collected and stored at −80°C. Blood cells were diluted with Incomplete RPMI Medium (IRM: RPMI 1640 with 25 mM HEPES, 11 mM glucose, 2 mM glutamine, 0.2% NaHCO₃, 0.2 mM hypoxanthine, 25 mg/l gentamicin, pH 7.4) and layered onto Lymphoprep™, centrifuged at 700 *g* for 25 min at RT, and the peripheral blood mononuclear cells were removed. From the remaining blood cells, the granulocytes were removed with plasmagel, and the infected erythrocytes (IE) at ring stage were cultured in Complete RPMI Medium (CRM: IRM with 0.5% Albumax II) at 2% haematocrit (HCT) and < 7% parasitaemia. Normal red blood cells were obtained from malaria-naïve volunteers and thoroughly washed in IRM. After 30–40 h, when developed into trophozoites expressing PfEMP1 on the IE surface, the IE were used for the binding assay. Immediately after processing, 50–100 μl of IE was also cryopreserved in glycerolyte (25 mM phosphate buffer pH 6.8, containing 57% glycerol, 16 g/l sodium lactate and 0.3 g/l KCL) and 50–100 μl resuspended and stored in TRIzol® at −80°C.

### Characterisation of HBMEC and HDMEC

To ensure that EC characteristics and receptor expression were maintained at higher number of passages and that receptor expression was comparable between HBMEC and HDMEC, the EC were characterised by flow cytometry. Confluent monolayers of cells were incubated with 10 μg/ml acetylated low-density lipoprotein, labelled with 1,1′-dioctadecyl-3,3,3′,3′-tetramethyl-indocarbocyanine perchlorate (Dil-Ac-LDL, Tebu) for 4 h at 37°C. Labelled cells were detached with Accutase®, washed with cold PBS/1% BSA/2 mM EDTA (P/B/E), and Dil-Ac-LDL was measured in the phycoerythrin channel of the flow cytometer. To determine receptor expression, cells were detached with Accutase®, washed with cold P/B/E, labelled with

conjugated antibody for 50 min at 4°C, washed with cold P/B/E and fixed in 2% paraformaldehyde prior to measurement on the flow cytometer. Antibodies used: FITC-conjugated mouse anti-human CD31, FITC-conjugated mouse anti-human CD36, PE-conjugated rat anti-human EPCR (all Biolegend) and APC-conjugated mouse anti-human ICAM-1 (BD).

### Binding to microvascular endothelial cells under flow

To minimise *var* gene switching (Scherf *et al*, 1998; Peters *et al*, 2007), the binding assay was performed, if possible, in the 1st developmental cycle; occasionally, assays were delayed to the 2nd or 3rd cycle. More than 90% of the UM and 75% of the CM cases were used in the 1st or 2nd cycle. The binding assay under flow conditions was performed using the Cellix microfluidics system (https://cellixltd.com/), as previously described (Madkhali *et al*, 2014). Briefly, HBMEC (Cell Systems, US) or HDMEC (Promocell, Germany) were cultured as per manufacturer's instructions and used up to passage 9, whilst retaining their endothelial characteristics as shown in Appendix Fig S1. The cells were stimulated overnight with 10 ng/ml TNF, dislodged with Accutase® and seeded in Vena8 biochips (Cellix) coated with 100 μg/ml fibronectin. Medium was changed every hour, and after 2–3 h, when cells formed a confluent monolayer, an IE suspension of 2% parasitaemia and 5% haematocrit in binding buffer (RPMI 1640 with 25 mM HEPES, 11 mM glucose, 2 mM glutamine, pH 7.2) was flowed through at shear stress of 0.4 dyne/cm² for 5 min at 37°C. A wash with binding buffer was performed for 7–9 min to remove unbound IE and bound IE were counted in 15 fields by microscopy using 200× magnification and the mean IE/mm² EC cell surface calculated. To determine the role of three endothelial receptors in binding of IE, the assay was carried out in the presence of 5 μg/ml αICAM-1 (clone 15.2, Serotec) or 5 μg/ml αCD36 (clone IV-C7, Sanquin, The Netherlands) antibody or 50 μg/ml recombinant EPCR (kind gift of Prof. M. Higgins, Oxford University). Percentage inhibition was calculated relative to binding in the absence of inhibitor and inhibition data were only used if binding in the absence of inhibitor was at least 20 IE/mm², to exclude binding variability by low levels of binding. This additional constraint on data collection resulted in a decreased number of observations for the inhibition data. The various conditions tested increased the duration of the assay and prevented us to measure replicates. Therefore, we counted 15 fields throughout the channel in the biochip. Most of the assays that resulted in binding data were performed in Malawi using fresh isolates (75% for the UM isolates and 62% for the CM isolates), but at a later stage, some of the assays were carried out in Liverpool from frozen parasites, which were shipped in a dry shipper to the UK.

### Determination of *var* gene transcripts

Infected erythrocytes at ring stage were stored in TRIzol® after processing the blood sample, and RNA was isolated by chloroform extraction and isopropanol precipitation, DNase treated (TURBO™ DNase, Ambion) and reverse transcribed (Tetro cDNA Synthesis Kit, Bioline) as previously described (Lavstsen *et al*, 2012). qPCR was carried out with SYBR Green PCR Master Mix (QuantiTect, Qiagen) with the *var* gene-specific primer set developed by the Lavstsen group consisting of 38 primer cocktails using the following cycling

## The paper explained

### Problem

Most malaria cases in Africa are the result of an infection with the human parasite *Plasmodium falciparum*, and the complications leading to severe malaria cause the majority of deaths. One syndrome of severe malaria, cerebral malaria (CM), is characterised by the accumulation of infected erythrocytes (IE) in the small blood vessels of the brain. The process by which IE bind to the endothelial cells lining the blood vessels is called cytoadherence, and over the years, several host receptors have been identified that are able to support the binding of IE via *P. falciparum* erythrocyte membrane protein 1 (PfEMP1). PfEMP1 is a highly variable protein consisting of different combinations of adhesive domains that provide a changing pattern of IE binding to host endothelium. This variation in adhesion potential has led to the hypothesis that the differential pathology seen in malaria infection (only 1–2% of the over 200 million cases will go on to develop severe disease) may be due to the different endothelial interactions mediated by PfEMP1 variants and specifically, for cerebral malaria, caused by the ability of IE to bind efficiently in the brain.

### Results

Several studies have attempted to correlate the binding phenotype of *P. falciparum* patient samples with clinical outcome with variable results due to the major technical challenges involved in this type of study. Key to understanding the role of adhesion in recruitment of IE to specific tissues such as the brain is the use of a relevant target for binding within a physiologically relevant assay system and the use of parasite samples as close to patient sampling as possible. To deliver this, we established flow-based adhesion assays to primary human microvascular endothelium, derived from brain and dermis, in our laboratories in Malawi and performed binding assays using patient-derived parasite samples from well-defined clinical paediatric cases with minimal *in vitro* expansion in culture. In parallel, we used molecular techniques to type the PfEMP1 variants involved in this adhesion.

Our results show that binding of IE from patients with CM to human brain microvascular endothelium (HBMEC) is higher than that seen with IE from patients with uncomplicated malaria (UM). However, when binding of CM- and UM-derived IE was examined to non-brain endothelial cells [human dermal microvascular endothelial cells (HDMEC)], the levels of binding were comparable, resulting in significant more binding of IE of UM patients to HDMEC compared to HBMEC. This suggests that in the majority of cases, represented by UM, *P. falciparum* avoids targeting the brain and that CM cases represent a subset of adhesion phenotypes that allow efficient binding to the cerebral vasculature. Our work represents one of the few direct strands of evidence directly linking the ability of IE to bind to brain endothelium with CM. In addition, the molecular typing has confirmed an important role for PfEMP1 variants with a binding signature for the host receptor endothelial protein C receptor in CM and implicated some novel PfEMP1 domains with potential associations with severe disease for further study.

### Impact

By understanding the pathways that contribute to the pathology of CM, we will be able to focus on a subset of binding variants on which to base the design of potential interventions. From our research, we now understand that binding of IE in the brain is an unusual property for *P. falciparum* parasites and that this behaviour is important in creating a local environment (rather than systemic) in the cerebral vasculature that causes brain swelling, the latter having previously been strongly associated with death from CM. Blocking or reversing IE adhesion in the brain through the design of vaccines to restricted PfEMP1 variants or receptor-based inhibitors could protect people from developing CM and post-CM neurological sequelae. Knowing that the local environment in the brain is important in disease also flags further study on what is happening at the sites of IE cytoadherence and how we might control the pathological processes operating there.

conditions: 95°C for 15 min; 40 cycles of 95°C for 30 s/50°C for 40 s/65°C for 50 s; and a dissociation step of 95°C for 1 min/55°C for 30 s/95°C for 30 s. Primers were validated with dilutions of 3D7 gDNA on our qPCR system (Agilent Technologies, Stratagene). Detailed description of the primers can be found in Mkumbaye *et al* (2017) and are summarised in Appendix Table S1, including the description of four additional primers. Levels of *var* transcripts were determined relative to the endogenous housekeeping genes *seryl-tRNA synthetase* and *aldolase* using the formulae $\Delta Ct$ *var*-primer = $Ct$ *var*-primer − $C$t average endogenous primers. The transcript unit (Tu) was calculated as Tu = $2^{(5-\Delta C_{t\,var-primer})}$ with any $\Delta C_t$ *var*-primer > 5, which is a low abundance transcript, assigned a value of 5, resulting in Tu = 1 (Lavstsen *et al*, 2012). qPCR data for which the melting temperature of the primer cocktail was out of the expected Tm range (Mkumbaye *et al*, 2017) were also assigned a Tu value of 1. A Tu value of 32 equates to equal transcript levels as the endogenous control genes.

The primers targeting DBLζ4 and DBLζ6 domains have overlapping specificities, and therefore, the highest Tu value of one of the primer sets was used when the total Tu value of DBLζ_all was calculated. Selected primer cocktails were used when RNA amounts were still scarce.

### Statistical analysis

Patient sample size prior to the study was calculated based on the cytoadherence data from Ochola *et al* (2011). Based on the difference in binding between CM and UM, a power of 80% and a significance of 5%, a sample number of 23 per group were calculated. The binding assays were performed by one person, who also counted the IE bound to the EC. This was done immediately after the wash-step (without fixation). *P*-values for differences in the binding assay were determined with an unpaired two-tailed *t*-test and for the qPCR and clinical data with the Mann–Whitney *U*-test for continuous variables and Fisher's exact test for categorical variables (Prism, version 5; GraphPad, USA). The correlation coefficient *rho* was calculated with two-tailed non-parametric Spearman test for cytoadherence data and clinical data (Stata, version 11; Stata-Corp, USA) and for cytoadherence data and qPCR data (Prism). To observe trends for the latter, correlations were not corrected for multiple comparisons and all correlations with a *P*-value ≤ 0.1 were included. This lenient cut-off was chosen to allow the inclusion of relatively high correlations of the inhibition binding data for CM isolates, for which fewer data were available and which failed to reach a *P*-value < 0.05 due to small sample size. To ensure biological significance of the correlations, only those in which at least 25% of the isolates had Tu values of ≥ 16 are included in Appendix Table S2. This cut-off value was also used by Mkumbaye to define a reasonable level of transcripts and facilitate the interpretation of the data (Mkumbaye *et al*, 2017).

**Expanded View** for this article is available online.

## Acknowledgements

We thank the children and their parents/guardians for participation in this study, Dr. Ian McCormick (Malawi-Liverpool-Wellcome Trust Clinical Research Programme and University of Liverpool) for his assistance in determination of retinopathy, the nurses and clinicians at the Paediatric

Research Ward (Malawi-Liverpool-Wellcome Trust Clinical Research Programme and Blantyre Malaria Project) for their clinical care of the children, Ida Nkhonjera (Malawi-Liverpool-Wellcome Trust Clinical Research Programme) for assistance in recruiting UM cases, Prof. Brian Faragher and James Dodd (Liverpool School of Tropical Medicine) for advice on statistical analysis, Prof. Matt Higgins (University of Oxford) for providing rEPCR and Prof. Malcolm Molyneux for critically reviewing the manuscript. This work was funded by a Wellcome Trust Investigator Award to AGC (grant 095507), Danish Council for Independent Research (grants 1333-00220 and DFF–4004-00624B) and US National Institutes of Health (NIH R01HL130678) (TK, CW, JJ) and internal support from the College of Osteopathic Medicine, Michigan State University (TT, KS).

## Author contributions

AGC, the principal investigator, developed the concept of the study, and JS was leading the study in Malawi. The study was designed by AGC, CAM, KBS and JS for patient recruitment; AGC, TS and JS for cytoadhesion experiments; and TL for *var* gene analysis. KBS, TET and PP recruited patients, provided clinical care and recorded clinical parameters. CAM, KBS and JS performed the patient data analysis. TS, JS, MM and NVC developed methodology and performed the cytoadhesion experiments, and JS analysed the obtained data. TL, CWW and JSJ developed the methodology for *var* gene analysis, and JS and MM performed the qPCR experiments. JS, JSJ, CWW and TL analysed the qPCR data. JS and AGC wrote the manuscript with input from TL, CAM, TET, KBS, JSJ and CWW.

## Conflict of interest

The authors declare that they have no conflict of interest.

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
