## [Review Process File · EMBO Molecular Medicine]

Cerebral malaria is associated with differential cytoadherence to brain endothelial cells

Janet Storm, Jakob S. Jespersen, Karl B. Seydel, Tadge Szeszak, Maurice Mbewe, Ngawina V. Chisala, Patricia Phula, Christian W. Wang, Terrie E. Taylor, Christopher A. Moxon, Thomas Lavstsen and Alister G. Craig

Review timeline:

Submission date:	25 March 2018
Editorial Decision:	30 April 2018
Revision received:	7 November 2018
Editorial Decision:	26 November 2018
Revision received:	11 December 2018
Accepted:	14 December 2018

Editor: Céline Carret

Transaction Report:

1st Editorial Decision

30 April 2018

Thank you for the submission of your manuscript to EMBO Molecular Medicine. We have now heard back from the three referees whom we asked to evaluate your manuscript.

As you will see from the reports below, all referees appreciate the potentials of the study being the 1st of its kind and the value of the clinical samples used. However the 3 of them highlight serious shortcomings in terms of methodology (cells used, growth of parasites w/o serum, var genes expression...), missing controls to give confidence in the results, unclear statistics, unexplored role of platelets, confusing structure and writing of the paper leaving the reader w/o a clear novel take home message.

We would like to give you the possibility to revise your article and are ready to extend the deadline to 6 months, should it facilitates things for you. We would like to highlight the points that would need revising for the paper to be further evaluated: re-do the adhesion part of the study using parasites grown in human serum (hoping that you do have frozen isolates), including repeating the adhesion inhibition assays to include appropriate controls, comparing adhesion with and w/o human serum in the binding assay medium. Other missing controls and fixing some methodological flaws should be done as well. Addressing the reviewers' concerns in full and experimentally as much as possible will be necessary for further considering the manuscript in our journal and this appears to require a lot of additional work and experimentation. I am unsure whether you will be able or willing to address those and return a revised manuscript within the 6 months deadline. I would understand if you were to rather decide to publish the manuscript rapidly and without any significant changes elsewhere.

Please note that EMBO Molecular Medicine strongly supports a single round of revision and that, as acceptance or rejection of the manuscript will depend on another round of review, your responses should be as complete as possible.

EMBO Molecular Medicine has a "scooping protection" policy, whereby similar findings that are published by others during review or revision are not a criterion for rejection. Should you decide to

submit a revised version, I do ask that you get in touch after three months if you have not completed it, to update us on the status.

Please also contact us as soon as possible if similar work is published elsewhere. If other work is published we may not be able to extend the revision period beyond six months.

I look forward to receiving your revised manuscript.

Should you find that the requested revisions are not feasible within the constraints outlined here and choose, therefore, to submit your paper elsewhere, we would welcome a message to this effect.

***** Reviewer's comments *****

Referee #1 (Comments on Novelty/Model System for Author):

The technical quality is mixed. The application of flow based adhesion assays using primary human brain ECs and highly characterised malaria clinical isolates is strength. The statistical analysis is flawed as discussed below.

Referee #1 (Remarks for Author):

In this manuscript, Storm et al. studied parasites isolated from the peripheral blood of children with cerebral malaria (CM), a deadly complication of *Plasmodium falciparum* infection. Brain sequestration is considered a hallmark of CM, but it is unknown how common it is in uncomplicated malaria (UM) infections. Here, the authors investigated whether patients with CM have different parasite binding types than children with uncomplicated malaria (UM). Strengths of the study are the combination of well-defined pediatric CM cohort, flow-based in vitro binding assays, and TNF-activated primary human brain microvascular endothelial cells (HBMEC). This is the first study of its type and revealed higher parasite-HBMEC binding levels in CM patients and suggests a partial role of both EPCR and ICAM-1 in HBMEC binding, which has been a matter of controversy and debate. Weaknesses are the manuscript is poorly written for a broader audience and many of the conclusions will be difficult for the uninitiated to follow. Additionally, important controls are missing for the in vitro binding experiments and there are significant limitations to the Spearman Correlations in Table 3. Overall, this is a unique dataset on an important disease complication, but several issues require attention.

Comments

1. Title is inaccurate. This study does not examine "...higher Pf-IEs in the brain". It shows that Pf-IEs isolated from peripheral blood of CM cases have higher in vitro binding capacity for primary HBMEC than UM cases.
2. Abstract is unclear. The sentence increased binding to HBMEC "was not significantly associated with EPCR and ICAM-1" is confusing. I think this emphasis is misplaced, since a major conclusion of this study is that HBMEC binding was partially mediated by EPCR and ICAM-1. The abstract sentence refers to no statistically significant difference in the EPCR and ICAM-1 binding dependence of the CM and UM isolates. However, there are many potential explanations including that similar parasites bound in either case. We can't know because the authors did not directly examine the adherent parasites, but it is well known that small subpopulations of parasites can be selected in these binding assays. I think it is more important to highlight the role of EPCR and ICAM-1 in mediating HBMEC binding, since this has been questioned (Ref #72 Asazi et al.).
3. Key controls are missing for the HBMEC/HDMEC binding assays. The authors assume that both EPCR and ICAM-1 are present on both cell types, but only HDMEC express CD36. It is critical to show FACS expression of EPCR, ICAM-1, and CD36 levels on both cell types to interpret the binding inhibition assay. If endothelial receptor levels differ, then this could result in different parasite types adhering to HBMEC and HDMEC.
4. The authors did not mention in methods how many times the binding of each isolate was replicated. If the authors couldn't replicate the experiments due to the low amount of parasite they

should clearly state it as a limitation. *P. falciparum* binding assays present an inherent variability that even increases in flow-based experiments. This variability might have account for the lack of significance in certain comparisons, for example Figure 2C.

5. It is a strange omission that the authors do not cite Kessler et al (Cell Host & Microbe 2017). The same DC8 and group A EPCR binding var transcripts were increased in this pediatric CM cohort (2013-2015 seasons) and in Kessler et al (2015-2016 seasons), further strengthening the association over multiple malaria seasons. In addition, Kessler et al showed that platelets might have an important role in brain swelling. It is interesting that the authors of this study also found an association between platelet levels and parasite cytoadhesion. Again, It would be important to mention that both studies showed platelet levels to be associated with cerebral malaria in multiple seasons and using different approaches

6. A novel aspect of this study is the attempt to link parasite var types in circulating blood to in vitro binding phenotypes (Spearman Correlations, Table 3). However, I have a number of concerns about this analysis.

- It does not appear the correlations were corrected for multiple comparisons. If not, p values will decrease or become statistically insignificant.
- Another concern is that this approach does not account that diverse groups of var genes can mediate similar binding properties. This has major implications for many of the binding comparisons.
- For instance, previous work suggests that both DC8 and Group A EPCR binding var types can adhere to HBMEC in vitro. Additionally, a variety of different ICAM-1 binding domains (group A or group B&C) would be expected to adhere well to TNF-activated HBMEC due to ICAM-1 upregulation. This creates considerable "noise" and may artificially weaken associations. This could contribute to why only the CIDRa1.6a domain was linked to HBMEC and HDMEC binding, and no other highly expressed var transcripts in peripheral blood.
- The same problem exists for ICAM-1 dependence. The issue is further complicated because PfEMP1 with dual EPCR + ICAM-1 versus dual CD36 + ICAM-1 may have different activity for HBMEC and HDMEC, especially if CD36 expression differs. On this same point, CIDRa1.6a linked to HBMEC and HDMEC binding is found in PfEMP1 with dual EPCR + ICAM-1 (DC4), yet does not appear for either cell type for ICAM-1 dependence. This hints at the possible complexity of parasites adhering via ICAM-1 and obscuring role of CIDRa1.6a. Without directly studying the adherent parasites for var expression, the authors must be more circumspect in their conclusions.

Of the 4 comparisons in Table 3, I think the only interpretable one is "EPCR-dependent binding" because grouped domains (CIDRa1.DC8, CIDRa1.A, and CIDRa_all) have the same binding properties. There is still the possibility of "interference" between DC8 and group A EPCR binders, but less problematic than the other 3 comparisons. Indeed, all three EPCR primer groups have decent correlations, suggesting that both DC8 and group A EPCR binders played a role in the EPCR-binding dependency.

My recommendation is to focus on EPCR-dependent binding, correct for multiple comparisons, and address study limitations in discussion. The authors also found some intriguing associations with C-terminal domains, but this section is very difficult to follow for the uninitiated and somewhat "hand-wavy". Alternatively, more sophisticated multivariate approaches must be used that can account for the variable var gene transcripts between patients. Or to directly sequence var transcripts from the parasites selected on HBMEC and HDMEC.

7. The observation that CM isolates adhere to brain ECs via EPCR and/or ICAM-1 is buried and deserves higher prominence. It has been questioned whether EPCR is a true endothelial receptor (Ref #72 Asazi et al.). This is the first study that has examined CM patient isolates on primary HBMEC under flow conditions.

Suggestions:

- Supplemental Figure 1 is a key figure and should be moved to the main manuscript. The paired analysis shows the raw IE binding and a significant difference with ICAM-1 and EPCR blockade. However, it is not clear why only a subset of parasite isolates from Fig. 2 is in FigS1. The set of evidence for both receptors should be better organised and presented.
- If possible, it would be helpful to show more of the raw Pf- IE binding data to better understand some of the exceptions in the blockade studies (e.g. parasites dependent on neither receptor).

- A number of groups have reported that EPCR only partially contributes to HBMEC binding (refs 36-38, 42, 82). Conversely, Azasi et al. (ref 72) argue that only DC8 Pf-IE adhere to EPCR, but completely lose HBMEC binding in the presence of human plasma/sera. In order to understand the EPCR-dependence reported in this study, it is essential to evaluate some CM parasite isolates for binding in the presence of human plasma/sera. This is a key experiment.

8. Care should be taken throughout the manuscript to better organize the flow of information and provide summaries along the way.

- More attention needs to be placed to guide the reader through the complicated var biology.

Currently, the intro refers the reader to a supplemental PfEMP1 schematic. In Result section, var primer findings are presented with no explanation of what they may mean.

- It is an important methodological detail that all parasites were normalized to 2% parasitemia for binding. This explanation should come at the beginning of the results, not the last paragraph.

- I found the rationale for the study difficult to follow because Intro brought up issues like whether "sequestration is a cause or effect of CM is still uncertain" or "whether cytoadherence is involved in the pathogenesis of CM". These seem like straw man arguments. Indeed, this group has been the leader in showing the retinal pathology and CM histology. I think it is clearer to say brain sequestration is hallmark of pediatric CM malaria, but the extent this occurs in non-CM cases is unknown. This study addresses whether CM cases have higher proportion of circulating Pf-IEs that can bind to HBMEC than UM cases.

Referee #2

(Comments on Novelty/Model System for Author):

The model used is good, and that is indeed the improvement from previous studies trying to address this question. The authors use human primary endothelial cells from different organs and determine binding of IE to different receptors of CM and non CM cases, as well as different var genes expression and their correlations.

Referee #2 (Remarks for Author):

The study by Storm and colleagues adds on previous work from some of the authors showing that IEs from CM patients bind EPCR and ICAM-1 more efficiently than IEs from UM.

In this new proposed article the authors use endothelial cells of brain and non-brain origin to replace the protein coated dishes used in the 2017 JID paper by Ndam et al, and the binding assay is now no longer static as previously reported but the authors now make use of a commercial microfluidic device.

The authors attempt to demonstrate that sequestration specific to the brain endothelium by certain PfEMP1 domains is the cause of cerebral malaria pathology.

This study will be of the interest of the malaria and pathology community as it should become the closest approximation to what happens in vivo. I point out below a few limitations of the study in its current version, as well as some aspects that in my view could improve the manuscript.

In figure 1 the authors show that the mean number of cytoadhering IEs to HBMEC from CM cases is higher than that of UM cases, but there is great overlap of distribution making it unclear if the medians comparison would also result in a significant difference. I would justify the analysis of means or medians, as throughout the different figures both are used.

It is not clear to me how the 2% parasitaemias were achieved, in the methods section the authors state that after plasmagel granulocytes were removed and parasitaemia obtained was 95%. Was this done with the straight from the arm blood? or after culture? To my knowledge plasmagel can only enrich that much for later forms than the ones found in circulation. I suggest clarifying.

The authors indicate that parasitaemias of UM cases were not determined on admission, but then also say that higher peripheral parasitaemias were selected to achieve a 2% parasitaemia for the cytoadherence assays, these are in my view inconsistent and should be revised.

Although the authors indicate that most samples were not grown for more than 3 cycles, I believe they should also state when was the analysis of var gene transcripts done, at the time of blood collection or after culture, when the adhesion study was performed?

The authors should also say how many samples from CM and UM cases were used for this analysis, were they the same as the 26 and 33 reported in Fig 1.

Would it be fair to say that CM samples simply have more PfEMP1 expressed than UM samples, regardless of which PfEMP1s are being expressed, or could it be that PfEMP1s which are not considered with the 38 set of primers could be expressed on the UM samples not determined? I suggest discussing it. I suggest including Supplementary figure A in the main text as it guides the reader to the different domain that were measured by RT-PCR and makes it easier to analyse the data in table 1.

Concerning figure 2 I think anti-CD36 could have served as a negative control on the HBMEC inhibition of binding data in Fig 2 to show that antibodies not targeting ligands which promote adhesion have no effect.

I wonder why are the sample sizes different in the different groups in the different analyses compared to figure 1? I recommend describing it further.

Table 3 is in my opinion not very easy to interpret it is not clear which data it includes. And it is not clear to me how many samples of each group, or if there was a minimal threshold of var expression to make the correlation? and for the number of binding IE/mm²? Without knowing that is, in my opinion, very difficult to understand how much is the contribution of samples that barely expressed var or did not bind. (On the table data is visible that for most samples the range goes down to 1.0 indicating low or no expression according to the methods section, were these also included?)

I recommend revising and maybe showing a few representative graphs of positive and negative correlations to guide the reader, as the ones shown in Supplementary figure 2.

I think the "All" data added on the CM and UM on table 3 brings an extra layer of complexity, without adding much to the results or discussion, so I would consider keeping it supplementary.

minor comment:

I would not refer to "recruitment" of IE to the brain, as in my opinion "recruitment" assumes that cells move towards either actively or in response to some gradient. What I believe the authors want to describe is that IE are retained in the brain once passing by. I suggest wording it differently.

Referee #3

This is a potentially fascinating and novel study examining whether *Plasmodium falciparum* isolates from patients with cerebral malaria show different adhesion characteristics (in terms of binding to primary brain- and skin-derived endothelial cells) compared to isolates from patients with uncomplicated malaria. var gene transcription profiles are also compared between the two groups (this aspect is not novel). The study contains interesting data, but is hard to interpret due to lack of clarity in some areas as described below. The authors should be congratulated on performing a challenging study with hard-to-obtain and clinically well-characterised parasite samples. It is a shame that there are some methodological flaws and that the data don't clearly provide substantial new insights.

Major concerns

1. Lack of a clear "bottom line".

It was difficult for the reader to work out exactly how these data move the field forward. There were no clear hypotheses being tested, and it wasn't clear how the experiments that were performed could differentiate between competing hypotheses and shed new light on the pathogenesis of cerebral malaria. This could be addressed by more clearly stating the hypotheses in the introduction, and describing how alternate outcomes would provide new insights into pathogenesis.

The abstract states that binding to brain microvascular endothelial cells "...was not significantly associated with binding to the endothelial receptors ICAM-1 or EPCR" but later states "higher levels of var gene transcripts predicted to bind EPCR and ICAM-1 were detected in CM isolates and for the EPCR-binding domains that correlated with EPCR-dependent binding to brain endothelial cells".

I'm sorry but this is really confusing and difficult to understand. These statements seem to contradict each other.

Is there alternative terminology to describe PfEMP1 so that confusing terms such as "var transcripts predicted to bind EPCR and ICAM-1" can be avoided? (Group A and B/A?) Also calling CIDRalpha1-containing variants "EPCR-binding PfEMP1" which is done throughout is confusing, given the ongoing questions about whether these variants really do bind EPCR when expressed on the IE surface (eg. ref 72)

The end of the abstract states "These data, for the first time, provide direct evidence for a mechanism of increased sequestration in the brain leading to CM." It is very unclear what this means. Please explain precisely what is the mechanism that has been uncovered here?

The discussion pg 7 states "Our main finding is that there are differential binding characteristics to brain endothelium between CM and UM patient isolates, particularly that the UM isolates do not bind as well to HBMEC as CM isolates."

If this is the major finding of this paper, unfortunately it is far from clear-cut (Fig 1). Yes, the mean binding level to HBMEC by CM isolates is higher than for UM isolates (110 IE/mm² vs 43 IE/mm², with a marginal p value of p=0.041), but there is extensive overlap between the two sets of samples. Many UM isolates bind to HBMEC better than many CM isolates. How should this be interpreted? For me, these data raise the question - why don't UM isolates cause cerebral sequestration and cerebral malaria, given that they bind to brain endothelium almost as well as CM isolates? The more striking finding is that UM isolates bind better to HDMEC than HBMEC (although again there is overlap), whereas CM isolates don't. So perhaps UM isolates don't bind in the brain because they bind better elsewhere, whereas CM isolates bind equally well anywhere, leading to substantial cerebral sequestration when parasite burdens are high?

I think the possible implications of the adhesion data could be explored in much more detail than they currently are in the manuscript.

2. Methodological flaws.

Unfortunately, there are some methodological flaws in the study with potential to have a serious impact on the validity of the data. It would be extremely beneficial if the authors could provide some reassurance that these methodological problems do not affect their results. Alternatively, some detailed discussion of the limitations of the work due to these problems is needed.

Lack of human serum in the parasite culture medium: it has been shown unequivocally that addition of human serum to culture medium during parasite maturation is essential for normal PfEMP1 expression on the IE surface (eg. see references below).

Frankland S. et al. Serum lipoproteins promote efficient presentation of the malaria virulence protein PfEMP1 at the erythrocyte surface. *Eukaryot Cell* 2007 6; 1584-1594

Ribacke U et al. Improved in vitro culture of *Plasmodium falciparum* permits establishment of clinical isolates with preserved multiplication, invasion and rosetting phenotypes. *PLoS One* 2013 8:e69781. doi: 10.1371/journal.pone.0069781.

Tilly AK et al. Type of in vitro cultivation influences cytoadhesion, knob structure, protein localization and transcriptome profile of *Plasmodium falciparum*. *Sci Rep* 2015 16;5:16766. doi: 10.1038/srep16766.

The presence of human serum has a huge effect on PfEMP1 expression (eg. see Fig 3 of Ribacke et al, comparing PfEMP1 surface expression in IE grown with either albumax or human serum). Yet according to the methods pg 12, parasites for the Storm et al. study were cultured in RPMI with 0.5% albumax II (i.e. without human serum). This is a serious technical flaw, given the importance of PfEMP1 in IE adhesion. The authors have substantially impaired their own ability to study PfEMP1-mediated adhesion by growing the parasites in albumax.

Can the authors justify why human serum was not included in the parasite culture medium? Given the essential requirement for human serum for adequate PfEMP1 expression, how meaningful are the data shown in this manuscript?

This manuscript already represents a huge amount of work, so I hesitate to suggest more experiments, but it may be the only way to validate the data. Would it be possible to repeat some

adhesion experiments with parasites grown in the presence of human serum to determine whether this does have a substantial impact on the adhesion data? (According to the methods, IE were cryopreserved, so this should be possible).

Endothelial cell line authentication: were the cell lines authenticated? The methods state that the primary cells were used up to passage 9. This is fairly standard procedure, but 9 passages still provide ample opportunity for minor contaminants in the primary cell preparation such as fibroblasts to out-grow the endothelial cells. Was there any validation of the "endothelial" nature of the cell line after say 5-6 passages? (eg. positive for expression of CD31, vWF and Dil-Ac-LDL uptake; negative for SMA; negative for CD36 (for HBMEC). This is important for interpretation of the data - were the cells being tested really endothelial cells?

Lack of negative controls for the adhesion inhibition experiments: there are no negative controls reported in any of the adhesion inhibition experiments, apart from a "no additive" sample. Surely, as a bare minimum, there should be an isotype control in the mAb experiments, and an irrelevant recombinant protein produced in the same system for the recombinant EPCR experiments? Why were controls not done? There seems to have been plenty of parasite material (some of it was frozen so there was excess material). And the Vena8 biochips have 8 channels. Why were controls (and indeed, technical replicates) not carried out?

Lack of serum in the adhesion assays: There is also no human serum included in the binding medium in the adhesion assays. A recent paper (ref 72, Azasi et al. PNAS 2018) shows that for a lab-adapted parasite line, EPCR-binding does not occur when normal human serum is included in the binding medium. Are the binding assays carried out by Storm et al. really physiologically-relevant in the absence of human serum?

3. Structure of the manuscript

The swapping between adhesion data and var gene profiling data made the manuscript hard to follow. The story would flow more logically if all the adhesion data were presented first (including the clinical correlates), then the var gene profiling results, and finally the correlations between var gene profiles and adhesion at the end. I found it hard to follow and understand the implications of the adhesion/var gene correlations, and the problem of multiple comparisons was not addressed.

The var gene profiling data overall confirms previous studies (which is useful), but doesn't really add anything new. There is inherent bias in the way these studies are done and reported, because almost all the primer sets used are designed to amplify group A and B/A var genes. Only a single primer set amplifying some group B and C var genes was used. So to then report the results as showing higher expression of various group A and B/A domains in CM versus UM is true, but can be misleading, implying that these are the only var genes groups that differ. Group B and C var genes might also differ between CM and UM isolates, but they haven't been studied in detail here. It would be useful to point this out to readers.

4. Other important issues:

Some important experimental details are missing, and it would help readers if fuller descriptions were given as outlined below.

Important methodological details missing for the primary endothelial cells: given that primary human brain endothelial cells (HBMEC) were used in this study, were all parasite isolates tested for adhesion on the same batch of primary HBMEC? Or were multiple different batches of primary cells from different donors used? If the primary cells were from different donors - how was variation in adhesion between donors accounted for?

The same question applied to the primary dermal endothelial cells.

Missing details for flow adhesion assays: methods pg 12 states "an IE suspension of 2% parasitaemia and 5% haematocrit (or equivalent) in binding medium..." How was a 2% suspension of IE achieved, given that the samples would vary in their starting parasitaemia? Presumably some dilution was carried out for samples with a parasitaemia >2%. How was this done? How was the parasitaemia checked after dilution? What about samples with a starting parasitaemia of <2%? Given that adhesion usually shows a strong positive correlation with number of IE added to an assay, it is essential that all parasite isolates were tested at the same parasitaemia and haematocrit.

What does "or equivalent" mean in the method described above? (This suggests that some samples were not at 2% parasitaemia and 5% haematocrit?)

Also, pg 7 states "after processing the UM blood samples, parasitaemia was determined semi-quantitatively by microscopy". How could accurate dilution above be done if the parasitaemia was only semi-quantitative?

"A wash with binding buffer was performed" (pg 13) - how long for?

"...bound IE counted in 15 fields...." was the counting done immediately after the wash?

Was the person counting IE adhesion blinded in regard to which samples were in which lanes of the vena8 biochip? This is important to avoid observer bias.

Under "Processing of whole blood" (methods pg 12), removal of PBMC with Lymphoprep and granulocytes with plasmagel are described, then it states "leaving >95% pure infected erythrocytes (IE)". This statement implies that the ring stage IE have been separate from uninfected RBC as well as from PBMC and granulocytes. Is that the case? If so, please describe in detail how the infected and uninfected RBC are separated from each other. If untrue, please rephrase the statement and clarify what is meant by "leaving >95% pure IE" (and explain what the remaining <5% of cells are).

Why was 0.4 dyne/cm² chosen as the shear stress to be used?

Minor concerns:

The title is very misleading - infected erythrocyte binding "in the brain" is not what is being reported here, it is binding to brain endothelial cells in vitro.

The abstract is confusing, hard to follow, and contains some odd statements. For example "... whether it (cerebral sequestration) is a major contributor to pathogenesis remains unclear." This statement does not stand up to scrutiny. Although there are many unanswered questions about cerebral malaria pathogenesis, the one thing that all recent (and no-so-recent) reviews on the topic would agree on is that IE sequestration in the brain is an essential component. Therefore, this comment in the abstract should be replaced.

Collection and storage of plasma and PBMC are described in detail in the methods pg 12, yet these items are not used in the study. Therefore, details on their collection could be removed?

1st Revision - authors' response

7 November 2018

We thank the reviewers for their comments and many constructive suggestions, which we believe have improved the manuscript. There are issues that we are unable to address, largely linked to our desire to perform the work in situ rather than remove the parasite samples and carry out the assays remotely. It is difficult to communicate just how challenging this study was to perform, and we thank the reviewers for recognising the massive amount of work that this manuscript represents.

We acknowledge (and discuss) some of the limitations and technical shortcomings, but we believe that these are minor and do not detract significantly from the robustness of our findings. The manuscript has been extensively rewritten based on the advice of the reviewers, particularly in focussing on the primary cytoadherence work and reducing the claims made for the secondary var gene analysis (although we think that these data are still interesting but agree that they are not definitive). We provide detailed responses to the reviewers' comments below:

***** Reviewer's comments *****

Referee #1 (Comments on Novelty/Model System for Author):

The technical quality is mixed. The application of flow based adhesion assays using primary human brain ECs and highly characterised malaria clinical isolates is strength. The statistical analysis is flawed as discussed below.

Referee #1 (Remarks for Author):

In this manuscript, Storm et al. studied parasites isolated from the peripheral blood of children with cerebral malaria (CM), a deadly complication of *Plasmodium falciparum* infection. Brain sequestration is considered a hallmark of CM, but it is unknown how common it is in uncomplicated malaria (UM) infections. Here, the authors investigated whether patients with CM have different parasite binding types than children with uncomplicated malaria (UM). Strengths of the study are the combination of well-defined pediatric CM cohort, flow-based in vitro binding assays, and TNF-activated primary human brain microvascular endothelial cells (HBMEC). This is the first study of its type and revealed higher parasite-HBMEC binding levels in CM patients and suggests a partial role of both EPCR and ICAM-1 in HBMEC binding, which has been a matter of controversy and debate.

Weaknesses are the manuscript is poorly written for a broader audience and many of the conclusions will be difficult for the uninitiated to follow. Additionally, important controls are missing for the in vitro binding experiments and there are significant limitations to the Spearman Correlations in Table 3. Overall, this is a unique dataset on an important disease complication, but several issues require attention.

Comments

1. Title is inaccurate. This study does not examine "...higher Pf-IEs in the brain". It shows that Pf-IEs isolated from peripheral blood of CM cases have higher in vitro binding capacity for primary HBMEC than UM cases.

We have changed the title so that it reflects more accurately our work. Being limited to 100 words including spaces is challenging.

2. Abstract is unclear. The sentence increased binding to HBMEC "was not significantly associated with EPCR and ICAM-1" is confusing. I think this emphasis is misplaced, since a major conclusion of this study is that HBMEC binding was partially mediated by EPCR and ICAM-1. The abstract sentence refers to no statistically significant difference in the EPCR and ICAM-1 binding dependence of the CM and UM isolates. However, there are many potential explanations including that similar parasites bound in either case. We can't know because the authors did not directly examine the adherent parasites, but it is well known that small subpopulations of parasites can be selected in these binding assays. I think it is more important to highlight the role of EPCR and ICAM-1 in mediating HBMEC binding, since this has been questioned (Ref #72 Asazi et al.).

We have rewritten the abstract. As mentioned by the reviewer, we have not examined the adherent parasites, we cannot select/collect them in our assay conditions. Based on the inhibition of binding by aICAM-1, we cannot conclude that ICAM-1 plays a major role in adhesion to HBMEC. For EPCR binding, we have unfortunately a fairly small sample size (due to availability of EPCR and parasite viability in 2015, as discussed). The paper has been re-written to reflect the positive support for EPCR and ICAM-1 in CM, but we have been careful not to overstep our data.

3. Key controls are missing for the HBMEC/HDMEC binding assays. The authors assume that both EPCR and ICAM-1 are present on both cell types, but only HDMEC express CD36. It is critical to show FACS expression of EPCR, ICAM-1, and CD36 levels on both cell types to interpret the binding inhibition assay. If endothelial receptor levels differ, then this could result in different parasite types adhering to HBMEC and HDMEC.

See the attached figure 1 (for reviewers) in which the expression levels of ICAM-1, EPCR and CD36 are shown at different passages. ICAM-1 and EPCR expression between HBMEC and HDMEC are comparable and also between the passage numbers (see also our comments to reviewer 3). As shown in the figure, CD36 is not detectable on HBMEC by flow cytometry.

4. The authors did not mention in methods how many times the binding of each isolate was replicated. If the authors couldn't replicate the experiments due to the low amount of parasite they should clearly state it as a limitation. *P. falciparum* binding assays present an inherent variability that even increases in flow-based experiments. This variability might have account for the lack of significance in certain comparisons, for example Figure 2C.

Besides the small amounts of IE (especially for CM isolates), the length of the assay was restrictive

to do more controls or do the assay in duplicate. In total there are 9 assay conditions (HBMEC: no inhibitor, α ICAM-1, rEPCR, α ICAM-1+rEPCR, HDMEC: no inhibitor, α ICAM-1, rEPCR, α ICAM-1+rEPCR and α CD36), which takes approximately 20 minutes per condition. To keep the EC viable after seeding in the biochips we never used more than 4 channels per biochip and had separate biochips for HBMEC and HDMEC. Therefore we were only able to do each condition once, but we increased the amount of fields counted per channel from 10 to 15. We have added text describing this limitation to the methods section.

5. It is a strange omission that the authors do not cite Kessler et al (Cell Host & Microbe 2017). The same DC8 and group A EPCR binding var transcripts were increased in this pediatric CM cohort (2013-2015 seasons) and in Kessler et al (2015-2016 seasons), further strengthening the association over multiple malaria seasons. In addition, Kessler et al showed that platelets might have an important role in brain swelling. It is interesting that the authors of this study also found an association between platelet levels and parasite cytoadhesion. Again, It would be important to mention that both studies showed platelet levels to be associated with cerebral malaria in multiple seasons and using different approaches

The Kessler et al paper has been added to the introduction and discussion sections and we can only apologise for its omission. The link with platelets is also interesting and could lead to several speculative suggestions, including the loss of parasite killing due to reduced platelet levels. This is a fascinating area, but our paper is unable to make much progress other than to report similar observations to other clinical studies on CM.

6. A novel aspect of this study is the attempt to link parasite var types in circulating blood to in vitro binding phenotypes (Spearman Correlations, Table 3). However, I have a number of concerns about this analysis.

- It does not appear the correlations were corrected for multiple comparisons. If not, p values will decrease or become statistically insignificant.

- Another concern is that this approach does not account that diverse groups of var genes can mediate similar binding properties. This has major implications for many of the binding comparisons.

- For instance, previous work suggests that both DC8 and Group A EPCR binding var types can adhere to HBMEC in vitro. Additionally, a variety of different ICAM-1 binding domains (group A or group B&C) would be expected to adhere well to TNF-activated HBMEC due to ICAM-1 upregulation. This creates considerable "noise" and may artificially weaken associations. This could contribute to why only the CIDRa1.6a domain was linked to HBMEC and HDMEC binding, and no other highly expressed var transcripts in peripheral blood.

- The same problem exists for ICAM-1 dependence. The issue is further complicated because PfEMP1 with dual EPCR + ICAM-1 versus dual CD36 + ICAM-1 may have different activity for HBMEC and HDMEC, especially if CD36 expression differs. On this same point, CIDRa1.6a linked to HBMEC and HDMEC binding is found in PfEMP1 with dual EPCR + ICAM-1 (DC4), yet does not appear for either cell type for ICAM-1 dependence. This hints at the possible complexity of parasites adhering via ICAM-1 and obscuring role of CIDRa1.6a. Without directly studying the adherent parasites for var expression, the authors must be more circumspect in their conclusions. Of the 4 comparisons in Table 3, I think the only interpretable one is "EPCR-dependent binding" because grouped domains (CIDRa1.DC8, CIDRa1.A, and CIDRa_all) have the same binding properties. There is still the possibility of "interference" between DC8 and group A EPCR binders, but less problematic than the other 3 comparisons. Indeed, all three EPCR primer groups have decent correlations, suggesting that both DC8 and group A EPCR binders played a role in the EPCR-binding dependency.

My recommendation is to focus on EPCR-dependent binding, correct for multiple comparisons, and address study limitations in discussion. The authors also found some intriguing associations with Cterminal domains, but this section is very difficult to follow for the uninitiated and somewhat "handwavy".

Alternatively, more sophisticated multivariate approaches must be used that can account for the variable var gene transcripts between patients. Or to directly sequence var transcripts from the parasites selected on HBMEC and HDMEC.

After correction of multiple comparisons none of the correlations were significant anymore, mainly due to the large number of qPCR assays conducted and our experimental design being focussed on the cytoadherence part. However, without the correction there are interesting correlations that can direct future research and we want to share these with the community. Therefore, we simplified table 3 and made it a supplementary table and limited the discussion of the results. It would be difficult to just focus on EPCR-dependent binding as this could be considered as 'cherry-picking' (and does little to progress the field as this correlation has already been very strongly demonstrated in multiple papers) and ignores the other potential correlations, such as those with the DBL ζ and DBL ϵ domains. We have added clear statements in the results and discussion sections about the lack of statistical power

7. The observation that CM isolates adhere to brain ECs via EPCR and/or ICAM-1 is buried and deserves higher prominence. It has been questioned whether EPCR is a true endothelial receptor (Ref #72 Asazi et al.). This is the first study that has examined CM patient isolates on primary HBMEC under flow conditions.

Suggestions:

- Supplemental Figure 1 is a key figure and should be moved to the main manuscript. The paired analysis shows the raw IE binding and a significant difference with ICAM-1 and EPCR blockade. However, it is not clear why only a subset of parasite isolates from Fig. 2 is in FigS1. The set of evidence for both receptors should be better organised and presented.

- If possible, it would be helpful to show more of the raw Pf- IE binding data to better understand some of the exceptions in the blockade studies (e.g. parasites dependent on neither receptor).

- A number of groups have reported that EPCR only partially contributes to HBMEC binding (refs 36-38, 42, 82). Conversely, Azasi et al. (ref 72) argue that only DC8 Pf-IE adhere to EPCR, but completely lose HBMEC binding in the presence of human plasma/sera. In order to understand the EPCR-dependence reported in this study, it is essential to evaluate some CM parasite isolates for binding in the presence of human plasma/sera. This is a key experiment.

- *We added supplementary figure 1 to the main text and is now figure 3A. The figure contains the same data as in figure 3 B-D (previously figure 2) and the impression of a smaller dataset is probably due to several of the lines lying on top of each other so it looks like there is less data.*

- *We will make all the raw data available. Supplementary figure 3 shows the inhibition of binding by ICAM-1 and rEPCR combined and we mention in the discussion that the binding of a number of isolates are not dependent on either receptor.*

- *Firstly, we have performed the binding of ITvar19 to HBMEC in the presence of 10% human serum and as shown in the figure (new supplementary figure 1) the binding of this EPCR-binding variant is not affected by 10% human serum in our assay conditions, unlike previously published work referred to by the reviewer. We also determined the binding of 2 CM isolates and 1 UM isolate to HBMEC and their binding is also not affected by the presence of 10% human serum. We discuss the results of these experiments in the main text.*

8. Care should be taken throughout the manuscript to better organize the flow of information and provide summaries along the way.

- More attention needs to be placed to guide the reader through the complicated var biology. Currently, the intro refers the reader to a supplemental PfEMP1 schematic. In Result section, var primer findings are presented with no explanation of what they may mean.

We have extended the description of the var genes in the introduction and added the PfEMP1 schematic to the main text (now figure 1) to aid the description of the results.

- It is an important methodological detail that all parasites were normalized to 2% parasitemia for binding. This explanation should come at the beginning of the results, not the last paragraph.

- I found the rationale for the study difficult to follow because Intro brought up issues like whether

"sequestration is a cause or effect of CM is still uncertain" or "whether cytoadherence is involved in the pathogenesis of CM". These seem like straw man arguments. Indeed, this group has been the leader in showing the retinal pathology and CM histology. I think it is clearer to say brain sequestration is hallmark of pediatric CM malaria, but the extent this occurs in non-CM cases is unknown. This study addresses whether CM cases have higher proportion of circulating Pf-IEs that can bind to HBMEC than UM cases.

- We added the normalisation to 2% parasitaemia and 5% HCT at the start of the results section.
 - We have changed the introduction to make the aim of the study clearer. There are some issues with simplifying CM pathology. While recent work has shown the potential for the EPCR/aPC/PAR1 axis to influence pathology, we are still some way from a unified theory or mechanism (indeed, we doubt this exists). One of the major criticisms of the murine CM model community is that the human CM field have presented cytoadherence as undisputable fact in its involvement in CM pathology, and while we and others have tried to build this case (including this manuscript), we still need to exercise caution in what we say. However, we have taken the reviewers' comments on board and attempted to simplify our arguments and highlight the strong associations that have been identified to improve the clarity of the paper.

Referee #2 (Comments on Novelty/Model System for Author):

The model used is good, and that is indeed the improvement from previous studies trying to address this question. The authors use human primary endothelial cells from different organs and determine binding of IE to different receptors of CM and non CM cases, as well as different var genes expression and their correlations.

Referee #2 (Remarks for Author):

The study by Storm and colleagues adds on previous work from some of the authors showing that IEs from CM patients bind EPCR and ICAM-1 more efficiently than IEs from UM.

In this new proposed article the authors use endothelial cells of brain and non-brain origin to replace the protein coated dishes used in the 2017 JID paper by Ndam et al, and the binding assay is now no longer static as previously reported but the authors now make use of a commercial microfluidic device.

The authors attempt to demonstrate that sequestration specific to the brain endothelium by certain PfEMP1 domains is the cause of cerebral malaria pathology. This study will be of the interest of the malaria and pathology community as it should become the closest approximation to what happens in vivo. I point out below a few limitations of the study in its current version, as well as some aspects that in my view could improve the manuscript. In figure 1 the authors show that the mean number of cytoadhering IEs to HBMEC from CM cases is higher than that of UM cases, but there is great overlap of distribution making it unclear if the medians comparison would also result in a significant difference. I would justify the analysis of means or medians, as throughout the different figures both are used.

The distribution in data is inherent to the use of patient isolates. We use the mean to analyse the cytoadherence data and this has been used in previous adherence assays, based on the distribution of the data.

It is not clear to me how the 2% parasitaemias were achieved, in the methods section the authors state that after plasmagel granulocytes were removed and parasitaemia obtained was 95%. Was this done with the straight from the arm blood? or after culture? To my knowledge plasmagel can only enrich that much for later forms than the ones found in circulation. I suggest clarifying.

We agree that the section in the methods is not clear. The plasmagel was used to remove the granulocytes (top fraction) from the blood sample and the bottom fraction then consist of a minimum of 95% of erythrocytes, not IE. We have rewritten this section to make it clearer.

The authors indicate that parasitaemias of UM cases were not determined on admission, but then also say that higher peripheral parasitaemias were selected to achieve a 2% parasitaemia for the cytoadherence assays, these are in my view inconsistent and should be revised.

For the UM cases there was only a grading based on parasites per high power field, as mentioned in the methods. The parasite density determined per WHO standard (parasites/white blood cells) was only done for the CM cases. This has been adjusted in the text. The 2% parasitaemia used for the cytoadherence assay was determined by counting 1000 RBC on a thin smear after culturing until the mature trophozoite stage (see also comment 4 of reviewer 3).

Although the authors indicate that most samples were not grown for more than 3 cycles, I believe they should also state when was the analysis of var gene transcripts done, at the time of blood collection or after culture, when the adhesion study was performed?

The authors should also say how many samples from CM and UM cases were used for this analysis, were they the same as the 26 and 33 reported in Fig 1.

We state in the methods that straight after processing the blood sample, the IE were resuspended and stored in Trizol. Thus the analysis of var genes was done directly after blood collection. The sentence at the start of the methods for determination of var transcripts has been changed to make it clearer.

Table 2 states the number of CM and UM cases per primer used for the var gene analysis. This is sometimes more than the amount of isolates used for the cytoadherence assay.

Would it be fair to say that CM samples simply have more PfEMP1 expressed than UM samples, regardless of which PfEMP1s are being expressed, or could it be that PfEMP1s which are not considered with the 38 set of primers could be expressed on the UM samples not determined? I suggest discussing it. I suggest including Supplementary figure A in the main text as it guides the reader to the different domain that were measured by RT-PCR and makes it easier to analyse the data in table 1.

- *We have added the PfEMP1 schematic to the main text (is now figure 1) to aid the description of the var gene analysis.*
- *We do discuss the limitations of using this primer set for the var gene analysis, especially the lack of good primers for CD36-binding CIDR domains group B and C var genes.*
- *As far as we know no one has determined the amount of PfEMP1 on patient isolates. From the literature there is also no evidence to support a strong correlation between the amount of PfEMP1 and its binding characteristics. Binding interactions between PfEMP1 and EC receptors are complex and do not correlate with the amount of PfEMP1 on the IE or amount of receptor on EC, but include several kinetic parameters. See also our reply to comment 2 from reviewer 3*

Concerning figure 2 I think anti-CD36 could have served as a negative control on the HBMEC inhibition of binding data in Fig2 to show that antibodies not targeting ligands which promote adhesion have no effect.

I wonder why are the sample sizes different in the different groups in the different analyses compared to figure 1? I recommend describing it further.

- *See also our reply to comment 4 of reviewer 1. Besides the small amounts of IE (especially for CM isolates), the length of the assay was restrictive, so we had to prioritise the inclusion (or not) of additional controls. In total there are 9 assay conditions (HBMEC: no inhibitor, α ICAM-1, rEPCR, α ICAM-1+rEPCR, HDMEC: no inhibitor, α ICAM-1, rEPCR, α ICAM-1+rEPCR and α CD36) which takes approximately 20 minutes per condition.*
- *For the inhibition data we used a threshold of 20 IE/mm² in the absence of inhibitor, hence the apparent differences in the sample sizes. This is described in the methods section.*

Table 3 is in my opinion not very easy to interpret it is not clear which data it includes. And it is not clear to me how many samples of each group, or if there was a minimal threshold of var expression to make the correlation? and for the number of binding IE/mm²? Without knowing that is, in my opinion, very difficult to understand how much is the contribution of samples that barely expressed var or did not bind. (On the table data is visible that for most samples the range goes down to 1.0 indicating low or no expression according to the methods section, were these also included?) I recommend revising and maybe showing a few representative graphs of positive and

negative correlations to guide the reader, as the ones shown in Supplementary figure 2. I think the "All" data added on the CM and UM on table 3 brings an extra layer of complexity, without adding much to the results or discussion, so I would consider keeping it supplementary.

We changed and simplified table 3 and made it a supplementary figure. The discussion of the results is also made clearer. Given the lack of power of this study to identify statistically significant associations, we do not believe that it would be useful to expend our analysis.

minor comment:

I would not refer to "recruitment" of IE to the brain, as in my opinion "recruitment" assumes that cells move towards either actively or in response to some gradient. What I believe the authors want to describe is that IE are retained in the brain once passing by. I suggest wording it differently.

We might dispute the use of the term "recruitment" as this is not necessarily an active process requiring movement within the vessel, but for clarity we have changed this. Adhesion has two major phases – recruitment from flow and firm attachment. IE need to be able to do both and the flow assay includes recruitment, unlike the static assay. This is not a critical issue here and the stationary binding measured in this work is a combination of both phases.

Referee #3 (Remarks for Author):

This is a potentially fascinating and novel study examining whether *Plasmodium falciparum* isolates from patients with cerebral malaria show different adhesion characteristics (in terms of binding to primary brain- and skin-derived endothelial cells) compared to isolates from patients with uncomplicated malaria. *var* gene transcription profiles are also compared between the two groups (this aspect is not novel). The study contains interesting data, but is hard to interpret due to lack of clarity in some areas as described below. The authors should be congratulated on performing a challenging study with hard-to-obtain and clinically well-characterised parasite samples. It is a shame that there are some methodological flaws and that the data don't clearly provide substantial new insights.

Major concerns

1. Lack of a clear "bottom line".

It was difficult for the reader to work out exactly how these data move the field forward. There were no clear hypotheses being tested, and it wasn't clear how the experiments that were performed could differentiate between competing hypotheses and shed new light on the pathogenesis of cerebral malaria. This could be addressed by more clearly stating the hypotheses in the introduction, and describing how alternate outcomes would provide new insights into pathogenesis.

We have rewritten some sections of the introduction and stated our aim more clearly. We have also re-ordered the text to make the cytoadherence work more prominent.

The abstract states that binding to brain microvascular endothelial cells "...was not significantly associated with binding to the endothelial receptors ICAM-1 or EPCR" but later states "higher levels of *var* gene transcripts predicted to bind EPCR and ICAM-1 were detected in CM isolates and for the EPCR-binding domains that correlated with EPCR-dependent binding to brain endothelial cells". I'm sorry but this is really confusing and difficult to understand. These statements seem to contradict each other.

*We are trying to make clear that determining the *var* gene expression is not the same as measuring actual binding of the isolates to EC. The significant higher levels of certain *var* transcripts (for example the ones containing domains predicted to bind EPCR or ICAM-1) does not mean the IE bind to EPCR or ICAM-1. We show that in our cytoadherence assay we do not detect a significant difference between ICAM-1 dependent binding of CM and UM isolates or in binding to HBMEC or HDMEC. For EPCR binding, we have unfortunately a fairly small sample size (due to availability of EPCR and parasite viability in 2015, as discussed). We have discussed our caution in interpreting our results earlier, but the point has been taken.*

Is there alternative terminology to describe PfEMP1 so that confusing terms such as "*var* transcripts

predicted to bind EPCR and ICAM-1" can be avoided? (Group A and B/A?) Also calling CIDRalpha1-containing variants "EPCR-binding PfEMP1" which is done throughout is confusing, given the ongoing questions about whether these variants really do bind EPCR when expressed on the IE surface (eg. ref 72)

We do not think it would be a good idea to use a new vocabulary, and have adopted the standard language for the PfEMP1 field. As the reviewer mentions, receptor binding by specific PfEMP1 variants are under discussion/scrutiny and therefore care has to be taken to describe var transcripts and their binding potential. We have rephrased the text where possible to clarify this.

The end of the abstract states "These data, for the first time, provide direct evidence for a mechanism of increased sequestration in the brain leading to CM." It is very unclear what this means. Please explain precisely what is the mechanism that has been uncovered here?

We have rewritten the abstract to make it more focussed. We have also removed the term "mechanism", as this is open to mis-interpretation.

The discussion pg 7 states "Our main finding is that there are differential binding characteristics to brain endothelium between CM and UM patient isolates, particularly that the UM isolates do not bind as well to HBMEC as CM isolates."

If this is the major finding of this paper, unfortunately it is far from clear-cut (Fig 1). Yes, the mean binding level to HBMEC by CM isolates is higher than for UM isolates (110 IE/mm² vs 43 IE/mm², with a marginal p value of p=0.041), but there is extensive overlap between the two sets of samples.

Many UM isolates bind to HBMEC better than many CM isolates. How should this be interpreted? For me, these data raise the question - why don't UM isolates cause cerebral sequestration and cerebral malaria, given that they bind to brain endothelium almost as well as CM isolates? The more striking finding is that UM isolates bind better to HDMEC than HBMEC (although again there is overlap), whereas CM isolates don't. So perhaps UM isolates don't bind in the brain because they bind better elsewhere, whereas CM isolates bind equally well anywhere, leading to substantial cerebral sequestration when parasite burdens are high?

I think the possible implications of the adhesion data could be explored in much more detail than they currently are in the manuscript.

There is variability in binding of the isolates to the endothelial cells, inherent to working with patient isolates. Indeed, almost any clinical study will show a similar distribution of data-points. We show significant higher binding of CM isolates to HBMEC compared to UM isolates and this relates to the UM isolates binding much better to HDMEC, as the reviewer mentions. We do discuss this binding phenotype of UM isolates. We have changed the order of the manuscript and focus more on the adhesion data and made the text clearer. One solution would be to show our data as bar charts, but we believe the use of dot plots provides clearer information, even if it might cause some confusion.

The issue about why there are overlaps between the categories in their adhesion levels is an interesting one, and probably indicates that binding is only part of the pathway to CM pathology. Having been accused of making things too complicated already, perhaps this is a discussion for another day.

2. Methodological flaws.

Unfortunately, there are some methodological flaws in the study with potential to have a serious impact on the validity of the data. It would be extremely beneficial if the authors could provide some reassurance that these methodological problems do not affect their results. Alternatively, some detailed discussion of the limitations of the work due to these problems is needed.

Lack of human serum in the parasite culture medium: it has been shown unequivocally that addition of human serum to culture medium during parasite maturation is essential for normal PfEMP1 expression on the IE surface (eg. see references below).

- Based on existing literature at the start of the project in 2012 (Frankland et al) we decided that the use of Albumax could be justified and thus we compared the binding of the laboratory isolate A4 grown in 0.5% Albumax and 10% human serum (see attached reviewer figure 2). There is no difference in adhesion in our assay conditions and by using Albumax, we could standardise the assay conditions. In addition, for pragmatic reasons, it also reduced the problems of sourcing and using AB negative human serum for parasite culture in Malawi.

- We thank the reviewer for the comprehensive list of manuscripts and below each cited paper is a summary of the findings. From this we do not see that there is evidence to support a strong correlation between the amount of PfEMP1 and its binding characteristics. Binding interactions between PfEMP1 and EC receptors are complex and do not correlate with the amount of PfEMP1 on the IE or amount of receptor on EC, but include several kinetic parameters. Therefore, we used a standardised procedure, including Albumax-containing medium, to measure binding to endothelium as the best assay available. We were not able to measure the amount of PfEMP1 on the IE.

Frankland S. et al. Serum lipoproteins promote efficient presentation of the malaria virulence protein PfEMP1 at the erythrocyte surface. *Eukaryot Cell* 2007 6; 1584-1594

From paper: PfEMP1 expression is much lower in Albumax: ~20-25% compared to IE grown in serum supplemented medium. But in the static binding assay to CD36 and ICAM-1 protein there is still 55-70% of binding left for IE grown in Albumax-medium compared to serum-medium and for CSA binding there is no difference. So the amount of PfEMP1 is not linear with receptor binding.

Ribacke U et al. Improved in vitro culture of Plasmodium falciparum permits establishment of clinical isolates with preserved multiplication, invasion and rosetting phenotypes. *PLoS One* 2013 8:e69781.doi: 10.1371/journal.pone.0069781.

Similar results as the manuscript above, less surface expression of PfEMP1 in Albumax-medium. But no binding assays are described in this paper.

Tilly AK et al. Type of in vitro cultivation influences cytoadhesion, knob structure, protein localization and transcriptome profile of Plasmodium falciparum. *Sci Rep* 2015 16;5:16766. doi:10.1038/srep16766.

Laboratory isolates 3D7 and FCR3 were cultivated in the presence of 0.5% Albumax or 10% human serum for at least three months. 3D7 IE, but not FCR3 IE, grown in Albumax-medium had fewer knobs than IE grown in serum-medium. PfEMP1 expression was not determined, but the determination of gene expression resulted in some var genes being upregulated in Albumax-medium and some in serum-medium. Static binding assays were done with CHO cells expressing receptors, with different results for 3D7 compared to FCR3. As the authors summarise, they did not observe a correlation between the presence and absence of knobs or gene expression profiles with cytoadhesion to various human endothelial receptors.

The presence of human serum has a huge effect on PfEMP1 expression (eg. see Fig 3 of Ribacke et al, comparing PfEMP1 surface expression in IE grown with either albumax or human serum). Yet according to the methods pg 12, parasites for the Storm et al. study were cultured in RPMI with 0.5% albumax II (i.e. without human serum). This is a serious technical flaw, given the importance of PfEMP1 in IE adhesion. The authors have substantially impaired their own ability to study PfEMP1-mediated adhesion by growing the parasites in albumax.

From the manuscripts above, this issue of using Albumax-containing medium is not as clear as the reviewer suggests. We do not know how much PfEMP1 is needed to bind to receptors on cells. And we also do not know how much surface expression of receptor is needed for binding. The kinetics of ligand-receptor binding are complicated and not properly addressed for PfEMP1 mediated binding. From our data on a lab isolate it is clear that in our assay conditions there is no difference in binding of IE, whether grown in serum- or Albumax-containing medium (see reviewer figure 2). In our experiments we used Albumax for all isolates and only cultivate the IE for a maximum of 6 days.

Can the authors justify why human serum was not included in the parasite culture medium?
Given the essential requirement for human serum for adequate PfEMP1 expression, how meaningful

are the data shown in this manuscript?

See above. We had not seen any differences in adhesion between human serum and Albumax grown parasites in our preliminary experiments, and using Albumax meant that we could standardise our conditions.

This manuscript already represents a huge amount of work, so I hesitate to suggest more experiments, but it may be the only way to validate the data. Would it be possible to repeat some adhesion experiments with parasites grown in the presence of human serum to determine whether this does have a substantial impact on the adhesion data? (According to the methods, IE were cryopreserved, so this should be possible).

As mentioned above, we have data on lab strain A4 grown in serum- or Albumax-containing medium, and there are no differences in binding to HBMEC or HDMEC (see reviewer figure 2).

Endothelial cell line authentication: were the cell lines authenticated? The methods state that the primary cells were used up to passage 9. This is fairly standard procedure, but 9 passages still provide ample opportunity for minor contaminants in the primary cell preparation such as fibroblasts to outgrow the endothelial cells. Was there any validation of the "endothelial" nature of the cell line after say 5-6 passages? (eg. positive for expression of CD31, vWF and Dil-Ac-LDL uptake; negative for SMA; negative for CD36 (for HBMEC). This is important for interpretation of the data - were the cells being tested really endothelial cells?

See also our reply to comment 3 from reviewer 1.

It is fair to say that standard protocols would restrict the use of primary endothelium up to passage 6-7. However, the high cost of HMBEC meant that we needed to extend this to passage 9. We characterised the EC at the start and at various stages during the project. We have now extended this in response to the reviewers' comments and the data are shown in attached reviewer figure 1. There is hardly any variation in the endothelial characteristics between the passage numbers, as measured by the uptake of Dil-Ac-LDL, CD31 expression and the upregulation of ICAM-1 expression by TNF. This result is also added as a sentence in the methods section. Receptor expression at the different passage numbers is also shown. CD36 is not detectable on HBMEC by flow cytometry.

Lack of negative controls for the adhesion inhibition experiments: there are no negative controls reported in any of the adhesion inhibition experiments, apart from a "no additive" sample. Surely, as a bare minimum, there should be an isotype control in the mAb experiments, and an irrelevant recombinant protein produced in the same system for the recombinant EPCR experiments? Why were controls not done? There seems to have been plenty of parasite material (some of it was frozen so there was excess material). And the Vena8 biochips have 8 channels. Why were controls (and indeed, technical replicates) not carried out?

See also our comments for reviewer 1 (point 4) and reviewer 2. In an ideal world these would be controls to consider. However, due to the relatively small amount of parasite material (not more than 200 µl IE from CM cases of which some was needed to freeze in glycerolyte and Trizol), choices on the assays to be performed had to be made. Besides the small amounts of IE, the length of the assay was also restrictive. In total there are 9 assay conditions (HBMEC: no inhibitor, αICAM-1, rEPCR, αICAM-1+rEPCR, HDMEC: no inhibitor, αICAM-1, rEPCR, αICAM-1+rEPCR and αCD36) which takes approximately 20 minutes per condition. To keep the EC viable after seeding in the biochips we never used more than 4 channels per biochip and had separate biochips for HBMEC and HDMEC. Therefore we were only able to do each condition once, but we increased the amount of fields counted per channel from 10 to 15. We added this limitation to the methods section.

NB. Previous cytoadherence work (albeit using laboratory isolates) has not shown any issues when isotype mAb or irrelevant proteins controls have been included.

Lack of serum in the adhesion assays: There is also no human serum included in the binding medium in the adhesion assays. A recent paper (ref 72, Azasi et al. PNAS 2018) shows that for a lab-adapted parasite line, EPCR-binding does not occur when normal human serum is included in

the binding medium. Are the binding assays carried out by Storm et al. really physiologically-relevant in the absence of human serum?

See also our reply to point 7 of reviewer 1.

Firstly, we have performed the binding of ITvar19 to HBMEC in the presence of 10% human serum and as shown in the figure (new supplementary figure 1) the binding of this EPCR-binding variant is not affected by 10% human serum in our assay conditions, unlike previously published work referred to by the reviewer.. We also determined the binding of 2 CM isolates and 1 UM isolate to HBMEC and their binding is also not affected by the presence of 10% human serum. Based on these results we decided that there was no need to test more isolates. We discuss the results of these experiments in the main text.

3. Structure of the manuscript

The swapping between adhesion data and var gene profiling data made the manuscript hard to follow. The story would flow more logically if all the adhesion data were presented first (including the clinical correlates), then the var gene profiling results, and finally the correlations between var gene profiles and adhesion at the end. I found it hard to follow and understand the implications of the adhesion/var gene correlations, and the problem of multiple comparisons was not addressed.

We agree and have changed the order of the results and also changed the correlation analysis. We changed and simplified table 3 and made it a supplementary figure. The discussion of the results has also been made clearer.

The var gene profiling data overall confirms previous studies (which is useful), but doesn't really add anything new. There is inherent bias in the way these studies are done and reported, because almost all the primer sets used are designed to amplify group A and B/A var genes. Only a single primer set amplifying some group B and C var genes was used. So to then report the results as showing higher expression of various group A and B/A domains in CM versus UM is true, but can be misleading, implying that these are the only var genes groups that differ. Group B and C var genes might also differ between CM and UM isolates, but they haven't been studied in detail here. It would be useful to point this out to readers.

We believe we have discussed the limitations of the var gene analysis with this primer set and agree with this reviewer that there are limitations.

4. Other important issues:

Some important experimental details are missing, and it would help readers if fuller descriptions were given as outlined below.

Important methodological details missing for the primary endothelial cells: given that primary human brain endothelial cells (HBMEC) were used in this study, were all parasite isolates tested for adhesion on the same batch of primary HBMEC? Or were multiple different batches of primary cells from different donors used? If the primary cells were from different donors - how was variation in adhesion between donors accounted for?

The same question applied to the primary dermal endothelial cells.

All experiments conducted with HBMEC were from the same batch. For HDMEC, several batches were used. However, we have examined the profile of the results obtained and we do not see any systematic bias linked to a batch. All batches of EC were tested by binding of a lab strain and showed no batch variation. For both HBMEC and HDMEC, a batch of cells is derived from one donor.

Missing details for flow adhesion assays: methods pg 12 states "an IE suspension of 2% parasitaemia and 5% haematocrit (or equivalent) in binding medium...." How was a 2% suspension of IE achieved, given that the samples would vary in their starting parasitaemia? Presumably some dilution was carried out for samples with a parasitaemia >2%. How was this done? How was the parasitaemia checked after dilution? What about samples with a starting parasitaemia of <2%?

Given that adhesion usually shows a strong positive correlation with number of IE added to an assay, it is essential that all parasite isolates were tested at the same parasitaemia and haematocrit.

What does "or equivalent" mean in the method described above? (This suggests that some samples were not at 2% parasitaemia and 5% haematocrit?)

Also, pg 7 states "after processing the UM blood samples, parasitaemia was determined semiquantitatively by microscopy". How could accurate dilution above be done if the parasitaemia was only semi-quantitative?

See also our response to a comment from reviewer 2. For the UM cases there was only a grading based on parasites per high power field, as mentioned in the methods. The parasite density determined per WHO standard (parasites/white blood cells) was only done for the CM cases. The 2% parasitaemia used for the cytoadherence assay was determined by counting 1000 RBC on a thin smear after culturing until the mature trophozoite stage. The text has been changed to describe this more clearly.

"A wash with binding buffer was performed" (pg 13) - how long for?
 "...bound IE counted in 15 fields...." was the counting done immediately after the wash?

We have added that the wash was for 7-9 minutes and that counting was done straight after the wash.

Was the person counting IE adhesion blinded in regard to which samples were in which lanes of the vena8 biochip? This is important to avoid observer bias.

No, one person was doing the assay and counted the IE bound straight after the wash (without fixation). Our team was too small to count the samples blinded, and we agree that this is not ideal.

Under "Processing of whole blood" (methods pg 12), removal of PBMC with Lymphoprep and granulocytes with plasmagel are described, then it states "leaving >95% pure infected erythrocytes (IE)". This statement implies that the ring stage IE have been separate from uninfected RBC as well as from PBMC and granulocytes. Is that the case? If so, please describe in detail how the infected and uninfected RBC are separated from each other. If untrue, please rephrase the statement and clarify what is meant by "leaving >95% pure IE" (and explain what the remaining <5% of cells are).

See also our reply to a comment from reviewer 2. We agree that the section in the methods is not clear. After lymphoprep to remove the PBMC, plasmagel was used to remove the granulocytes (top fraction) from the blood sample and the bottom fraction then consisted of a minimum of 95% of erythrocytes, not IE. We have rewritten this section. We cannot assure that all the granulocytes are removed by this method, hence the 95%.

Why was 0.4 dyne/cm² chosen as the shear stress to be used?

This is the shear stress in microvenules and used routinely in flow adhesion experiments.

Minor concerns:

The title is very misleading - infected erythrocyte binding "in the brain" is not what is being reported here, it is binding to brain endothelial cells in vitro.

We have changed the title to reflect the work more accurately. Being limited to 100 words including spaces is challenging.

The abstract is confusing, hard to follow, and contains some odd statements. For example "... whether it (cerebral sequestration) is a major contributor to pathogenesis remains unclear." This statement does not stand up to scrutiny. Although there are many unanswered questions about cerebral malaria pathogenesis, the one thing that all recent (and no-so-recent) reviews on the topic would agree on is that IE sequestration in the brain is an essential component. Therefore, this comment in the abstract should be replaced.

We have changed the abstract.

Collection and storage of plasma and PBMC are described in detail in the methods pg 12, yet these items are not used in the study. Therefore, details on their collection could be removed?

We have changed the text in the methods section to reflect this.

(Figures for reviewers from the next page)

Figures for reviewers

Reviewer Figure 1. Characterisation of HBMEC and HDMEC at different passages and expression of ICAM-1, EPCR and CD36

A) Dil-Ac-LDL uptake

HBMEC

Grey filled: unlabelled
Blue: p5
Red: p8
Green: p9

HDMEC

Grey filled: unlabelled
Blue: p4
Red: p6
Green: p8

Confluent monolayers of cells were incubated with 10 $\mu\text{g/ml}$ Acetylated Low Density Lipoprotein, labelled with 1,1'-dioctadecyl - 3,3,3',3'-tetramethyl-indocarbocyanine perchlorate (Dil-Ac-LDL, Tebu) for 4 hours at 37 $^{\circ}\text{C}$. Labelled cells were detached with Accutase, washed with PBS/1% BSA/2mM EDTA (P/B/E) and Dil-Ac-LDL was measured in the phycoerythrin channel of the flow cytometer. The grey peak is unlabelled cells and the blue, red and green peaks are labelled cells.

B) CD31 expression

HBMEC

Grey filled: unstained
Blue: p5
Bright green: p7
Red: p8
Green: p9

HDMEC

Grey filled: unstained
Blue: p4
Red: p6
Bright green: p7
Green: p8

Cells were detached with Accutase, washed with P/B/E and CD31 expression was detected with a FITC-conjugated mouse anti-human CD31 antibody (Biolegend) in P/B/E. The grey peak is unlabelled cells and the blue, red, bright green and green peaks are CD31-expressing cells.

C) Upregulation of ICAM-1 expression after TNF treatment

HBMEC

Dotted line: no TNF
Continuous line: + TNF
Blue: p5
Red: p8
Green: p9

HDMEC

Dotted line: no TNF
Continuous line: + TNF
Blue: p4
Red: p6
Green: p8

Cells were either not treated or treated with 10 ng/ml TNF for 16 hours, detached, washed with P/B/E and ICAM-1 expression was detected with a APC-conjugated mouse anti-human ICAM-1 antibody (BD) in P/B/E. The dotted lines are the cells without TNF, while the continuous lines are cells treated with TNF.

From the flow cytometry data shown, it can be concluded that HBMEC and HDEMEC retain their microvascular endothelial characteristics up to passage 9.

D) EPCR expression

HBMEC

Grey filled: unstained
Blue: p5
Red: p8
Green: p9

HDMEC

Grey filled: unlabelled
Blue: p4
Red: p6
Green: p8

EPCR expression of TNF treated cells was detected with a PE-conjugated rat anti-human EPCR antibody (Biolegend) in P/B/E.

E) CD36 expression

HBMEC

Grey filled: unstained
Blue: p5
Red: p8
Green: p9

HDMEC

CD36 expression of TNF treated cells was detected with a FITC-conjugated mouse anti-human CD36 antibody (Biolegend) in P/B/E.

From the flow cytometry data shown, it can be concluded that TNF-activated HBMEC and HDMEC have similar expression levels of ICAM-1 and EPCR. CD36 is not detectable on HBMEC but expressed on HDMEC, albeit not at very high levels.

Reviewer Figure 2. Cytoadherence of A4 grown in serum- or Albumax-containing medium.

The A4 *P. falciparum* strain was cultured in either RPMI medium containing 10% human serum (S) or 0.5% Albumax (A). Binding to HBMEC and HDMEC was determined under flow conditions and the number of IE bound per mm² EC surface was measured. Shown is the mean \pm SEM of at least 7 independent experiments. P-value was calculated by 2-tailed t-test.

In our assay conditions we do not see a significant difference in binding between A4 grown in serum-containing or Albumax-containing medium.

Thank you for the submission of your revised manuscript to EMBO Molecular Medicine. We have now received the enclosed reports from the referees that were asked to re-assess it. As you will see the reviewers are now globally supportive and I am pleased to inform you that we will be able to accept your manuscript pending minor editorial amendments.

In addition, please address the last requests of referees 1,2 and 3. They suggest adding data provided in your rebuttal letter to Appendix and modifying statistics.

I look forward to reading a new revised version of your manuscript as soon as possible.

***** Reviewer's comments *****

Referee #1 (Comments on Novelty/Model System for Author):

First time that parasite isolates from well-characterized CM cohort were studied for binding to TNF-activated primary HBMEC and HDMEC.

Referee #1 (Remarks for Author):

The revised manuscript from Storm et al. is much improved and addresses my comments. This was a challenging study to conduct. It tested many parasite isolates under flow conditions and with many permutations per isolate. I agree with the authors that the use of albumax-containing media to grow the parasites does not nullify the binding they observed in this study. A major strength of this study is that they performed flow-based adhesion studies on TNF-activated primary brain microvascular endothelial cells. Thus, they have taken great care to study binding under more physiological conditions than previous work. This work is novel and provides new molecular insights into the parasite adhesion types in cerebral malaria cases, as well as evidence for a role of EPCR and ICAM-1 in parasite adhesion.

Minor comment

1) I recommend that the FACS histograms provided for the reviewers showing CD36, EPCR and ICAM-1 expression on HBMEC and HDMEC cells should be included as a supplemental figure. This information is needed to interpret the antibody blockade data, and it is plausible that differences in receptor expression or levels could influence the importance of that receptor in parasite adhesion.

Referee #2 (Comments on Novelty/Model System for Author):

The used model is good, and that remains the improvement from previous studies addressing this question. The authors use human primary endothelial cells from different organs and determine binding of IE to different receptors of CM and non CM cases, as well as different var genes expression and their correlations.

Referee #2 (Remarks for Author):

The authors have answered in a satisfactory way to most of my questions, and I believe that the manuscript can be accepted after making a few minor adjustments, which at the statistical point of view I consider still very important to make the proposed.

I think the standard error of the mean is not well applied to the analyses the authors make and recommend changing to 95% CI or standard deviation. And I don't think that the fact that mean and SEM have been used before in analysing adherence assays serves as a justification to continue to do so.

I suggest that number of samples analysed the statistical test applied should be mentioned in all figure legends, which the authors do for most but not all figures, some have only the P value but not the test used.

Overall, I think the authors made the methods and result section more clear and improved the reading of the table with the var expression content, and I believe the malaria community and particularly the severe malaria aficionados will enjoy reading the manuscript.

Referee #3 (Remarks for Author):

This manuscript is greatly improved following revision by the authors. The aims and results are now clear, and the technical limitations are well described.

This is the first study to examine whether parasite isolates from patients with strictly-defined cerebral malaria differ in cytoadherence characteristics compared to parasite isolates from uncomplicated malaria patients, using physiologically relevant endothelial cell lines and flow conditions. It represents an enormous amount of work carried out under challenging conditions.

The data shown are clear, interesting and well-described. The additional data shown in the manuscript and the rebuttal letter resolve the major concerns noted in the initial round of review. I would suggest that the authors include the characterisation of the endothelial cells that is shown in the rebuttal letter as a supplementary figure in the manuscript.

An additional suggestion is that it would also be very useful to show graphs of the correlations between binding characteristics and clinical factors (such as parasite density and platelet count) as a supplementary figure (the correlation coefficients are reported in the text, but the actual data are not shown).

Overall, this manuscript describes novel data that will make an important contribution to the literature on cerebral malaria.

2nd Revision - authors' response

11 December 2018

We thank the referees for their kind words and their recommendation to accept the manuscript. We address their comments below.

Referee #1 (Remarks for Author):

The revised manuscript from Storm et al. is much improved and addresses my comments. This was a challenging study to conduct. It tested many parasite isolates under flow conditions and with many permutations per isolate. I agree with the authors that the use of albumax-containing media to grow the parasites does not nullify the binding they observed in this study. A major strength of this study is that they performed flow-based adhesion studies on TNF-activated primary brain microvascular endothelial cells. Thus, they have taken great care to study binding under more physiological conditions than previous work. This work is novel and provides new molecular insights into the parasite adhesion types in cerebral malaria cases, as well as evidence for a role of EPCR and ICAM-1 in parasite adhesion.

Minor comment

1) I recommend that the Facs histograms provided for the reviewers showing CD36, EPCR and ICAM-1 expression on HBMEC and HDMEC cells should be included as a supplemental figure. This information is needed to interpret the antibody blockade data, and it is plausible that differences in receptor expression or levels could influence the importance of that receptor in parasite adhesion.

We have now included the HBMEC and HDMEC characterisation by flow cytometry as Appendix figure S1 and added the methodology to the methods section.

Referee #2 (Remarks for Author):

The authors have answered in a satisfactory way to most of my questions, and I believe that the manuscript can be accepted after making a few minor adjustments, which at the statistical point of view I consider still very important to make the proposed.

I think the standard error of the mean is not well applied to the analyses the authors make and recommend changing to 95% CI or standard deviation. And i don't think that the fact that mean and SEM has been used before in analysing adherence assays serves as a justification to continue to do so.

We have changed the analysis of the cytoadherence data to mean \pm 95% CI. This is now in the graphs and mentioned in the results.

I suggest that number of samples analysed the statistical test applied should be mentioned in all figure legends, which the authors do for most but not all figures, some have only the P value but not the test used.

We apologise for the oversight. All figures and tables display now the number of samples, P-values and statistical test used.

Overall, I think the authors made the methods and result section more clear and improved the reading of the table with the var expression content, and I believe the malaria community and particularly the severe malaria aficionados will enjoy reading the manuscript.

Referee #3 (Remarks for Author):

This manuscript is greatly improved following revision by the authors. The aims and results are now clear, and the technical limitations are well described.

This is the first study to examine whether parasite isolates from patients with strictly-defined cerebral malaria differ in cytoadherence characteristics compared to parasite isolates from uncomplicated malaria patients, using physiologically relevant endothelial cell lines and flow conditions. It represents an enormous amount of work carried out under challenging conditions.

The data shown are clear, interesting and well-described. The additional data shown in the manuscript and the rebuttal letter resolve the major concerns noted in the initial round of review. I would suggest that the authors include the characterisation of the endothelial cells that is shown in the rebuttal letter as a supplementary figure in the manuscript.

We have now included the HBMEC and HDMEC characterisation by flow cytometry as Appendix figure S1 and added the methodology to the methods section.

An additional suggestion is that it would also be very useful to show graphs of the correlations between binding characteristics and clinical factors (such as parasite density and platelet count) as a supplementary figure (the correlation coefficients are reported in the text, but the actual data are not shown).

We think that showing the graphs of the correlation between binding and clinical parameters in addition to reporting these values in the results section would not add to the observation, which is not part of the main focus of the paper. Therefore we decided not to include these graphs in the Appendix.

Overall, this manuscript describes novel data that will make an important contribution to the literature on cerebral malaria.

Corresponding Author Name: Alister G. Craig and Janet Storm

Manuscript Number: EMM-2018-09164-V2